



# Simplified representation of runoff attenuation features within analysis of the hydrological performance of a natural flood management scheme

Peter Metcalfe[1*], Keith Beven[1,2], Barry Hankin[3] and Rob Lamb[4,1]

Correspondence to p.metcalfe@lancs.ac.uk

[1] Lancaster Environment Centre, Lancaster University, Lancaster, LA14YQ, UK
[2] Department of Earth Sciences, Uppsala University, Uppsala 75263, Sweden
[3] JBA Consulting, Sankey Street, Warrington, Cheshire WA1 1NN, UK
[4] JBA Trust, South Barn, Broughton Hall, Skipton, North Yorkshire, BD23 3AE, UK

**Abstract.** Hillslope Runoff Attenuation Features (RAFs) are soft-engineered overland flow interception structures utilised in natural flood management, designed to reduce connectivity between fast overland flow pathways and the channel. The performance of distributed networks of these features is poorly understood. Extensive schemes can potentially retain large quantities of runoff storage but there are suggestions that much of their effectiveness can be attributed to desynchronisation of subcatchment flood waves, and that inappropriately-sited measures may increase rather than mitigate flood risk. Fully-

distributed hydrodynamic models have been applied in limited studies but introduce computational complexity. The longer run-times of such models also restricts their use for uncertainty estimation or evaluation of the many potential configurations and storm sequences that may influence the timing and magnitude of flood waves.

We applied a simplified overland flow routing module and representation of RAFs to the headwaters of a large rural catchment in Cumbria, U.K., where the use of an extensive network of such features is proposed as a flood mitigation

strategy. The model was run in a Monte Carlo framework over a two-month period of extreme flood events which occurred in late 2015 that caused significant damage in areas downstream. Using the GLUE uncertainty estimation framework, we scored our set of acceptable realisations and these weighted behavioural realisations were rerun with one of three drain-down time or residence time parameters applied across the network of RAFs.

The study demonstrates that the impacts of schemes comprising widely-distributed ensembles of RAFs can be modelled

effectively within such a reduced complexity framework. It shows the importance of effective residence times on antecedent conditions in a sequence of events. We discuss uncertainties and limitations introduced by the simplified representation of the overland flow routing and RAF representation and how it could be verified and improved using experimental evidence. We suggest ways in which features could be grouped more strategically and means by which the synchronisation issue could be addressed.



# 1 Introduction

A catchment-based flood risk management (CBFRM) approach is becoming widely adopted (Werritty, 2006; Pitt, 2008; Dadson et al., 2017). Its principle is that storm runoff can be managed most effectively with a combination of catchment scale measures and downstream flood defences (Lane, 2017). A variety of CBFRM, often referred to as Natural Flood Management (NFM), or Working with Natural Processes (WwNP: EA, 2014), is an approach that utilises soft-engineered structures and interventions that both utilise and enhance the natural processes within the catchment (Calder and Alywood, 2006; SEPA, 2016; Lane, 2017). It is argued that NFM is a low-cost, scalable, approach that, in addition to improved flood resilience, can yield considerable benefits in terms of improved ecosystem services and stakeholder engagement (Lane et al., 2011).

This study arises from a project that was undertaken for the UK Rivers Trust intended to provide better understanding of how NFM could be applied strategically to the headwaters of three catchments in Cumbria, UK (Hankin et al., 2016). Measures included large-scale tree-planting to increase evaporation losses and to improve soil structure, and restoration of peat and heath to increase surface roughness. It included consideration of the installation on the hillslopes of a widely distributed network of run-off attenuation features (RAFs) to intercept fast overland flow Each intervention was modelled separately, allowing the effects of each to be examined.

The work builds on the approach piloted by Hankin et al. (2017) in the Eden, a rural catchment in Cumbria, UK with an area of 2248 km² upstream of Carlisle. There was significant flooding in settlements around the middle and lower reaches in 2005, 2009, and during Storm Desmond during 5th – 6th December, 2015. A high resolution, 2D inundation model based on solution of the Shallow Water Equations, (JFLOW, Lamb et al., 2009, Environment Agency, 2013) was applied to assess potential sites for distributed measures and understand relative impacts on the hydrograph, in terms of downstream benefits or damages avoided. This model was driven by multiple rainfall event sets incorporating spatial joint probabilities observed in the extremes from time series of observed rainfalls around the catchment (Lamb et al., 2010; Keef et al., 2013). It showed the potential for a widely-distributed NFM approach, but did not undertake detailed modelling of the proposed interventions, or test the model against real rainfall and discharge data. The responses of headwater catchments have been identified as having a disproportionate influence on the overall downstream flood risk (Pattinson et al., 2014), and were selected for further examination in this study. The strategic screening stage combined the JFLOW analysis with a catchment partner workshop order to identify areas with the greatest opportunities for NFM.

## 1.1 Aims and objectives

There is considerable uncertainty in both predictions of runoff response (Beven, 2006) and the effect of application of distributed flood mitigation measures, particularly in terms of their effects on effective hydrological parameters (Dadson, 2017; Hankin et al., 2017; Lane, 2017). Additional uncertainty will be introduced by the lack of knowledge of the effects of RAFs on the effective hydrological parameters, their response over series of storm events and hydraulic characteristics




whilst filling and draining. The response may be complicated by spatial variation in rainfall, and the effect of the sequencing of storm events on antecedent conditions and runoff generation.

The effectiveness of natural and distributed flood management schemes has been attributed, in part, to the desynchronisation of subcatchment flood waves (Thomas and Nisbet, 2007; Blanc et al., 2012). This suggests the possibility that

inappropriately-sited interventions could have a detrimental impact on the storm response by synchronising previously asynchronous waves, an impact that will be dependent on the configuration of subcatchments at different catchment scales. Given this, and the large combination of potential configurations and varieties of storm events, a pragmatic approach to evaluating the impacts of NFM would be "experimental" modelling such as proposed by Hankin et al. (2017), whereby many possible realisations of the catchment model and event sets are generated. The primary goal of the project was to deliver a

computationally efficient runoff model and representation of RAFs that would allow such an approach to be undertaken in reasonable timescales. This would necessarily would involve some simplifications of the RAF responses, as a fully hydrodynamic treatment will introduce considerable complexity and much increased run-times to any modelling exercise A semi-distributed hillslope runoff model simplified the simulation of storm runoff and allowed multiple runs over large scales with reasonable run times. An efficient representation of features was sought that would allow modelling of their

introduction without incurring significant computational cost. The approach is applied within the semi-distributed Dynamic TOPMODEL framework (Beven and Freer, 2001; Metcalfe et al., 2015, 2016), which aggregates hydrologically similar areas together whilst maintaining hydrological connectivity, provides a means to achieve this.

One objective of the project was to develop a representation for RAFs that incorporates sufficient information to adequately reflect the relevant aspects hydraulic response of the features as they fill and drain, including consideration of overflow

characteristics. A new, storage-based overland flow routing algorithm was utilised that could be modified to take into account interception of runoff by RAFs. It maintains a record of water levels in RAF which can be used to examine the drain down, filling and possible overflow of these features during the course of storm events.

Metcalfe et al. (2017) investigated the impact of the installation of barriers with varying underside clearance within the channel network. They observed that applying small clearances led to the filling of storage capacity in the course of a

double-peaked storm events. Features with larger clearances, although able to recover capacity more quickly between events, had less of an impact in intermediate storms. The project aimed to determine whether for hillslope interventions there is a similar trade-off between fast-draining features that retain the intense rainfall but operate less effectively over events of longer duration, and less permeable designs that retain more of the runoff but can become overwhelmed in larger storms. This effect was investigated by application of different levels of "leakiness" through the walls.

The expectation is currently that NFM will have an impact on small to moderate scale events. However, to be a practical strategy, it will be required to operate effectively across extreme events, but there is as yet little evidence of what is required to have an effective impact in these conditions. The application therefore assimilated actual rainfall data from multiple gauges for a period containing a sequence of flood events, including Storm Desmond, one of the largest events recorded in the UK, to modelling of the RAF intervention.



## 1.2   Runoff Attenuation Features as a Natural Flood Management technique

A wide range of measures are employed in catchment-based approaches to flood mitigation. These are intended to reduce hillslope – channel connectivity, slow surface and channel velocities and thus to mitigate the effects of fast runoff Techniques employed in NFM are reviewed by Quinn et al. (2013), EA (2014), SEPA (2016), Dadson et al., (2017) and

Lane (2017) and will not be discussed in detail here. Structures commonly found in NFM schemes include wooden barriers or debris dams in ephemeral channels, earth bunds, and ground scrapes or ponds. These are designed to intercept and store saturation overland or overbank flow and can be effective in disconnecting fast surface flow pathways from the channel and so increase hillslope storage. The walls of these structures are commonly constructed to be permeable or "leaky", reducing hydraulic stresses and allowing the stored runoff to drain out slowly. Another strategy is to have more impermeable walls but

to allow the storage to drain to the channel via a pipe. Other strategies, such as tree planting in critical locations, are aimed at encouraging runoff to infiltrate and follow slower subsurface pathways downslope. Their aim is to delay the arrival of runoff at the channel until after the main flood wave has passed, such that the downstream storm peak is attenuated.

Ghmire et al. (2014) simulated a single hillslope pond of capacity 27000m³ in a 74 km² catchment and showed that it could reduce peak flows by 9% in a 1 in 2 year event. The capacity of RAF features is generally much smaller than this, and in the

UK constrained by legislation that limits their size to 10000 m³ above which significant legal responsibilities are imposed (Wilkinson et al., 2010b). The storage required for a significant mitigating effect on storm flows is large, however. Metcalfe et al. (2017) showed that within a 29km² catchment 168,000m³ of hydrodynamic storage would be required to attenuate peak flow in order prevent flooding in a 1 in 75 year event.  A scheme of a realistic scale could therefore involve installation of many hundreds of RAFs, although the number required might be reduced by applying other measures.

Pattinson et al. (2014) examined the interacting effects of the Eden subcatchment flood waves using data for a large flood event in Carlisle in 2005 and concluded that their timing and magnitude predicted the majority of the variance in modelled downstream flood peaks. It will therefore be necessary to design such installations with some care. Slowing runoff that would have contributed to the hydrograph _before_ the peak could have the effect of increasing the peak magnitude. This is called the synchronicity problem. Peak timings vary, however, with the pattern and timing of rainfalls and antecedent

wetness in the catchment and the way in which the hydrographs from different subcatchments interact. Thus, it will be necessary to test the sensitivity of a design before implementation using an appropriate model of runoff generation and mitigation measures. This might, in itself, require many model runs that reflect different event characteristics and patterns, storage and drainage characteristics of RAFs.



## 2 Modelling strategies to evaluate the effect of RAFs

### 2.1 Runoff modelling

To assess the effects on the storm runoff of emplacement of RAFs it is necessary to predict the hillslope runoff contributions to the stream channel and the routing of the flood hydrograph in the channel network. Before the implementation of any RAFs the predicted catchment outlet discharges will allow calibration of model parameters against observed flows. Simulations with identical input data using representations of unaltered and modified catchments are then undertaken and the results of each compared. There are many runoff models that could be applied to this problem. The representation of small-scale RAF features would appear to require a fully-distributed, high resolution modelling approach. Such models are often highly-parameterised, with long run-times, making their use in uncertainty estimation frameworks challenging. To overcome such difficulties, in this study an implementation of the semi-distributed Dynamic TOPMODEL (Beven and Freer, 2001a; Metcalfe et al., 2015, 2016) is employed. The model has been applied in many studies (see, for example, Liu et al., 2009; Page et al. 2007; Metcalfe et al., 2017). The new implementation has demonstrated robustness to spatial and temporal discretisation applied to a 3.5km² upland catchment (Metcalfe et al., 2015). It was used to evaluate structures within the channel network of a small agricultural catchment in North Yorkshire, UK (Metcalfe et al., 2017) where an NFM scheme is proposed. This represented the in-channel barriers applied within a spatially-explicit network and hydraulic routing scheme, with the pattern of spatial runoff predicted by the hillslope component of Dynamic TOPMODEL.

The model extends TOPMODEL (Beven and Kirkby, 1979). The principles of the later version are detailed by, amongst others, Beven and Freer (2001a) and Metcalfe et al. (2015). The basic approach is the aggregation of "similar" landscape areas into Hydrological Response Units (HRUs) that are treated during the course of a simulation as having similar hydrological responses, based on common model parameters. The units may be of arbitrary size and not necessarily spatially contiguous, although they, along with their internal states, can be mapped back into space. This "discretisation" approach significantly reduces the complexity of the landscape model whilst retaining hydrological connectivity of the hillslope. The improved subsurface routing algorithm introduced in Dynamic TOPMODEL allows a more flexible approach to aggregation of catchment areas. Of particular relevance in the current context is the ability to collect areas identified with RAF interventions into one or more HRUs. In realisations reflecting unaltered catchments these units behave identically to surrounding landscape areas. To simulate the effect of applying one or more RAFs the surface runoff routing through the corresponding units can be altered to reflect their reduced connectivity with the hillslope. Model parameters are shown in Table 1.

**[INSERT Table 1]**

Once a HRU discretisation has been defined using relevant spatial overlays the model is run against rainfall and potential evapotranspiration data for a specified time period. For that period, it produces a time series of simulated discharges at the catchment outlet and time series of the internal states of each of the HRUs.





## 2.2 Overland flow routing in Dynamic TOPMODEL

Hunter et al. (2007) suggest that in some situations simplified, but physically-based, surface flow models can perform as well as a fully hydrodynamic formulations such as the Shallow Water Equations. In TOPMODEL and the first version of Dynamic TOPMODEL (Beven and Freer, 2001a) a network width approach was taken to routing surface flow (see Beven, 2012). In the implementation described by Metcalfe et al. (2015) a semi-distributed, storage-based surface flow routing module was introduced. This uses a routing scheme similar to that applied to the subsurface. Saturation excess from upslope HRUs is routed to downslope units by a surface flow distribution matrix $\boldsymbol{W_{of}}$ derived from the surface topography:

$$\boldsymbol{W_{of}} = \begin{pmatrix} 1 & 0 & \cdots & 0 \\ r_1 & p_{11} & \cdots & p_{1n} \\ \vdots & \vdots & \ddots & \vdots \\ r_n & p_{n1} & \cdots & p_{nn} \end{pmatrix} \qquad r_i + \sum_{j=1}^{n} p_{ij} = 1 \qquad (1)$$

Each row gives the proportions of the corresponding unit's flow that is directed to other units. For example, $p_{ij}$ is the proportion of unit $i$'s flow that is directed to unit $j$ and $p_{ii}$ is the proportion that remains within unit $i$. The vector $\boldsymbol{r}$ represents a lumped "river" unit such that $r_j$ is the proportion of downslope flux entering the channel network from unit number $j$. With an extended matrix a multi-reach river unit can also be defined. The matrix approximates transfer of flux between the different landscape units by averaging the intercell slopes of the elevation raster between cells falling into each of the landscape categories.

An assumption of a linear storage-discharge relationship is now made, whereby the discharge overland per unit contour out of a unit is proportional to depth of flow $d$. This implies a uniform velocity profile, so that the specific discharge per unit contour length from each HRU is $q_{out} = v_{of}d$, with $v_{of}$ its mean overland wave velocity. It can be shown (Metcalfe et al., 2015) that this leads to a coupled series of ordinary differential equations for the vector of discharges $\boldsymbol{q_{out}}$ from all of the HRUs

$$\boldsymbol{q_{out}} = \boldsymbol{A'd} \qquad (2)$$

where $\boldsymbol{A'} = \boldsymbol{v_{of}}^T(\boldsymbol{A} - \boldsymbol{I})$ and $\boldsymbol{A} = diag\left(\frac{1}{a}\right)\boldsymbol{W_{of}^T}diag(\boldsymbol{a}) - \boldsymbol{I}$, $\boldsymbol{d}$ the vector of average water depths and $\boldsymbol{a}$ the vector of areas for each of the HRUs. This system can be solved analytically by the so-called Eigenvalue method (Dummit, 2012). The storage distributed downslope is calculated to the end of the simulation time interval and any runoff routed to the channel is routed to the outlet using the network width approach. Surface excess storage redistributed to other units across the interval are added to the rainfall input for those units in the next time step. This novel approach allows for possible re-infiltration as run-on given a soil moisture deficit in downslope units. NFM measures such as tree shelter-belts that improve soil structure to enhance infiltration (e.g. Caroll et al., 2004) may have this effect in moderate events. Any excess over the saturated storage capacity of the profile, is treated as the equivalent quantity of overland flow and routed in the same fashion.



## 2.3    Approaches to modelling of RAFs

Physically-based models in general employ a gridded digital terrain model (DTM) to represent the surface. Features can be introduced into the landscape representation within the model by raising cells on their boundary to represent wooden walls or bunds. A fully-distributed hydrodynamic model, TUFLOW, was applied using this approach across the 74km² Tarland Burn catchment in Aberdeenshire, Scotland (Ghmire et al., 2010). In the Eden, Hankin et al. (2017) simulated hillslope ponds by deepening the appropriate cells in a 2m DEM and applied designed rainfall events to the JFLOW model. Both approaches may be sufficient to represent ponds and impermeable bunds but it will be difficult to account dynamically for infiltration, evaporation and losses through permeable walls, and thus the areas will be unable to drain down during the course of a simulation. The method will have limitations when applied to multiple storm events where recovery of storage capacity during recession periods is likely have an effect.

Hydraulic models of individual structures could be achieved by applying analogies to engineered interventions whose characteristics are better understood. Metcalfe et al. (2017) modelled the effects of the wooden channel barriers by analogy with underflow sluices employed in in irrigation schemes. Chow (1959) describes analytical storage-discharge relationships for such structures that utilise empirically-determined parameters. RAFs constructed of spaced timber members could be modelled by the Kirschmer-Mosonyi formula for flow through trash screens (Mosonyi, 1966), such as used in the intakes to power plants and waste-water treatment works. Overflow of these structures running out of capacity during the course of a storm event could be modelled analytically as though across a weir, another well-studied structure.

Wilkinson et al. (2010b) used a lumped representation of RAF storage. In this model a series of offline ponds representing the additional storage are connected to a single river reach. Flood discharge is routed through these treated as linear stores. The approach was applied in the Belford Burn catchment and suggested that 20000m³ of additional storage could have sufficient attenuation to prevent flooding in the smallest storm that would have caused damage. The storage was assumed to always be fully-utilised, but Metcalfe at al. (2017) showed that the interactions of a RAFs and the complex routing of flood waves down a realistic channel network meant that the storage associated with a scheme could be substantially underutilised, , even during large events Conversely, they can run out of capacity too soon, before or in the rising limb of these storms, to mitigate the flood peak sufficiently.

It has been suggested that RAFs are most effectively applied to reduce hillslope connectivity with the channel by placement across fast surface runoff pathways (Wilkinson et al., 2010a; Quinn et al., 2013). Such pathways can be identified by observation during storms or by examination of debris after events. Opportunities may also be determined through application of a hydrodynamic runoff model to identify accumulation areas. In the study case, the surface runoff for a designed storm of a given return period were modelled by using JFLOW applied across a 2m x 2m DTM. Areas with maximum surface storage exceeding a given depth were identified with sites for potential RAFs. An example map of intervention areas identified with this technique applied within the Eden is given in Figure 1.

**[INSERT FIGURE 1]**



## 2.4    Modelling RAFs with Dynamic TOPMODEL

Multiple-spatially distributed RAFs with similar characteristics can be lumped into individual response units. Characteristics could include height and permeability, position on the hillslope, upslope area, slope and proximity to the channel. Even if many such aspects are identified and result in multiple groupings, this approach will substantially reduce computational

overheads against a fully-spatially distributed representation.

Drainage is an important aspect of a RAF's behaviour. If operational the ground behind the RAF is likely to be saturated and little storage will infiltrate to the subsurface. There may still be evapotranspiration losses, which will occur at the maximum potential rate from the open surface, but this is likely to be minimal, especially during and between winter events. Direct drainage from the RAF could be via a pipe which, ignoring friction, gives a dependence of discharge on the square of the

hydrostatic head. Leaky wooden structures or permeable bunds may instead drain as though through a porous medium. In this case it be more realistic to relate output discharge linearly with the head (Beven, 2012). The structure could also be impermeable so that storage is lost downslope only when the features starts overflowing.

Overflow characteristics will also vary between features. In deepened hollows and ground scrapes any overflow in excess of the storage capacity will leave the feature in a similar direction and velocity as across the unmodified hillslope. In both leaky

and impermeable structures, assuming they are designed to maximise interception by following the local contours, the directions are again likely to be similar. Overflow rates over the feature will, however, depend on its construction, but in general can be represented by a form of non-contracted Weir equation (see Chow, 1959; Brater and King, 1976). If $q_{over}$ is the specific overflow and $h$ the water surface height above the weir crest then

$$q_{over} = C_d h^{1.5}$$ (3)

where $C_d$ is a coefficient that reflects the energy loss across the over-flowing edge of the structure and the associated

hydraulic jump as the flow becomes critical.

For wide, smooth bunds a broad-crested weir might be the best representation, where critical flow occurs at some point across the edge. In this case $C_d$ is a function of the weir breadth and the upstream head. Brater and King (1976) tabulate empirically determined values of $C_d$ against weir breadth and head for broad-crested weirs. Screens or barriers could more realistically be modelled as sharp-crested weirs, where the critical section occurs in the free drop outside the structure. In this

case the discharge coefficient can be calculated from the ratio of the crest height to the water depth $h_c$ (Chow, 1959):

$$C_d = 3.27 + 0.4 \frac{h}{h_c}$$ (4)

If the barrier is constructed of rounded members such as natural timber the value of $C_d$ calculated above increases proportionally with their radius (Jones, 1917). There are other functional forms when the weir discharge is submerged, outlined by Brater and King (1976).

The semi-distributed surface routing algorithm outlined in Section 2.2 is now applied to model groupings of RAFs. Units

associated with a grouping of RAFs are modified so that a smaller proportion of the downslope flux is directed to units than





for the unaltered landscape representation, and thus the feature fills when there is a net input of runoff. For features that are partly excavated or a bunded surface depression there will be an initial priming storage that must be filled before water starts draining downhill over the surface. This requires as an additional HRU parameter $ex_{min}$. Any surface excess storage below the surrounding ground surface (i.e. $ex_s < ex_{min}$) stays in the same unit and the remainder overflows across the surrounding hillslope.

The proportion of water lost from HRUs corresponding to permeable RAFs is emulated by altering the corresponding row in the surface distribution matrix [1] to reflect the "leakiness" of the walls. A factor $\Lambda \in [0,1]$ is defined as the proportion of flux draining downslope out of a RAF HRU unit. A modified routing matrix is developed where the elements of row $i$ control the flux distribution out of an aggregated RAF identified with unit $i$. In the case of a leaky dam on a hillslope the sum of the elements excluding the diagonal is $\sigma_i = r_i + \sum_{j=1 \cdots n, j \neq i}^{n} p_{,i,j}$. Setting $p_{i,i} = 1 - \Lambda$ and scaling the other elements of the $i$-th row by the factor $\Lambda/\sigma_i$ the row again adds to 1, as required. For a channel screen or woody debris dam the drainage direction is likely to be towards the nearest channel. For unit $i$ the river element $r_i$ of $\boldsymbol{W_{over}}$ is set to $\Lambda$, the diagonal component $p_{i,i}$ set to $1 - \Lambda$ and all the other $p_{,i,j}$ set to zero.

Note that only overland flow, represented by saturated excess, is routed out of the RAF HRU by the modified distribution matrix. Subsurface routing beneath the structure is unaltered. Each unit in effect behaves for the duration of a time step as a linear store with residence time $T_{res}(\Lambda, v_{of,i}) = \frac{1}{\Lambda v_{of,i}}$. For a roughly rectangular area storage could be considered as proportional to water depth. A parameter $ex_{max}$ [L] can be introduced to control the maximum storage; if $ex > ex_{max}$ the structure starts to overflow. An additional overflow matrix $\boldsymbol{W_{over}}$ is now defined to direct excess water out of the lumped features. For ground scrapes and bunded depressions this is identical with the unaltered surface distribution matrix $\boldsymbol{W_{surf}}$.

## 3   Uncertainty estimation framework

There is significant uncertainty in predictions of the spatial distribution and quantities of runoff generation (Beven, 2006, 2009, 2012). Attempts to assess the effectiveness of NFM will compare predictions for unaltered and modified catchments thus introduce even further uncertainty. In the case of the RAF interventions uncertainty will be introduced by, for example, feature location and their geometry and discharge characteristics. In addition, reduction in capacity through sedimentation or damage due to loading will mean that their performance may be non-stationary.

Uncertainty estimation and sensitivity analysis can provide a realistic assessment of the reliability of predictions of the impacts of NFM. These techniques will, however, require the generation of thousands or even millions of model realisations in which parameters are sampled from prior distributions of feasible values. With continued growth in computing power it is now feasible to run the fully-distributed JFLOW model over 750km² with a 2m resolution grid (175 million cells) in approximately real-time (Hankin et al., 2017). Metcalfe et al. (2017) were, through the use of parallel processing technology,



able to produce around 2000 realisations of storm routing with various channel and floodplain configurations and roughness. The introduction of in-channel interventions simulated with a hydraulic model increased run-times significantly and they were able to run only a limited sensitivity analysis of different configurations and dimensions. Thus, while high-performance computing may be fast enough for assessment for a selection of features and hillslope properties, it may still be inadequate

for uncertainty analysis across larger catchments and wide-scale NFM schemes comprising the large number of features required to meet the storage requirements to significantly attenuate real flood events.

A traditional approach to runoff modelling assumes a single landscape realisation, but it has been observed in many studies that different parameterisations can lead to very similar results, or equifinality (Beven, 2006). The approach taken by the project is a form of the Generalised Likelihood Uncertainty Estimation methodology (GLUE: Binley and Beven 1992; Beven

and Binley, 2014) employed in many studies (e.g. Beven and Freer, 2001b; Beven and Blazkova, 2004; Liu et al., 2009). GLUE accepts the possibility of many model realisations that can fit the observed behaviour. In the course of a calibration or other modelling exercise those realisations whose outputs meet an acceptability threshold with respect to some type of likelihood score calculated for each are retained; the score thenceforth being maintained alongside the corresponding realisation.

Typically, a large number of model parameters are sampled with a Monte Carlo approach from one or more prior distributions derived from, possibly subjective, knowledge of acceptable or likely ranges. These are then applied in turn to the system model and a simulation performed, resulting in multiple model realisations. Each is scored with a likelihood that takes as parameters relevant observables of the simulation, or performance metrics derived from it and observational data. Only those simulations achieving an acceptability threshold are retained yielding a "behavioural" parameter set. This gives a

posterior distribution of model realisations, each associated with a likelihood, or weighting, from which distributions of predicted variables can be obtained.

A triangular weighting function is commonly applied to the weightings for individual observations, whereby the value is unity at the likeliest value of the acceptability interval, zero at either limit, and linearly interpolated between these points. A trapezoidal form (e.g. Blazkova and Beven, 2009) could also be applied, whereby the value is unity between two threshold

values and linearly interpolated outside these to the acceptability criteria. The individual weights are combined, possibly with weighting applied to one or more measures, and normalised to produce a value between 0 and 1 (Beven and Freer, 2001b; Beven, 2006; Beven and Blazkova, 2009; Liu et al., 2009). All of the predicted values of the observables must lie within acceptable ranges or the realisation is rejected, even if the overall sum of weights is non-zero.

A suggested approach, applied in the case study described, for incorporating the uncertainty framework into evaluation of

RAF interventions is shown in Figure 2. Behavioural model realisations for the unaltered catchment model are obtained from a large number of randomly chosen model runs. Subsequently one or more modifications due to the applications of RAF interventions are applied to each of the base realisations. For each altered realisation the weighting score can be carried through from the baseline case, or if there is available information on the likelihoods of the effects of the interventions a score for the modified realisation can be calculated and combined with the weighting of the associated behavioural model.



**[INSERT FIGURE 2]**

## 4 Case study

### 4.1 Study catchment and calibration period

The Eden headwater catchments modelled in the study cover its area draining from the source near the border of Cumbria and North Yorkshire to the flow gauge, EA number 760101, at Great Musgrave Bridge (2.363234 W, 54.5126 N), a drainage area of 223km² (Figure 3). The catchment is 55.4% acid, improved or rough grassland and 36.0% bog or scrub and heath. Bedrock geology is Permian and Triassic sandstones lain on Carboniferous limestones and there is some influence of groundwater pathways (Ockenden and Chappell, 2010). Tree cover is minimal, comprising just 2.5% of its area, though there has been significant recent tree planting that is not yet thought to have had any major effect on flood peaks. Overall annual rainfall is in the region of 1200mm, but there is a strong synoptic and orographic influence (EA, 2009).

In early November 2015 a south-westerly airflow became established that brought warm moisture-laden air from subtropical regions. This was followed by a period of exceptional storm events that included Storms Abigail (15th-16th November), Barney (18th November) and Desmond (5th-6th December). The final extra-tropical cyclone caused significant damage and over 2000 homes were flooded at Carlisle, further downstream. A record 1680m³/s discharge was recorded at 9am on December 6th at Sheepmount Weir (54.905332 N, 2.952091 W) in Carlisle. The town of Appleby (54.578719 N, 2.488839 W), was also badly affected by this event; a peak discharge of 372 m³ was recorded at 18:00 on the 5th of December at the UK Environment Agency gauge at Great Musgrave Bridge a few kilometres upstream.

**[INSERT FIGURE 3]**

The early autumn period of 2015 was unusually dry and soil water deficits were in October more than 10mm greater than the long term average (Marsh et al., 2016). The calibration period chosen begins at the end of this month when the soil moisture deficit was at its peak and ends in the recession period of Storm Desmond. Processed 15 minute time series of rated discharges were obtained for gauge 760101. There is one other flow gauge within the catchment, at Kirkby Stephen (see Figure 3). Tipping-bucket recorder (TBR) rainfall data at 15 minute intervals were provided by the EA and a set of these gauges lying within 10 km of the catchment were identified. Given the extreme rainfall, some gauges went off-line during the storms and were removed from the set, leaving four gauges with complete records over the calibration period from which a rainfall record was interpolated. This met the water balance to within 5% and applied to the entire catchment area.

Although significant events, Storms Abigail and Barney did not cause damaging flooding in this catchment. This suggests that a reduction of the 7.0 mm/h peak of Storm Desmond to that of Abigail, at 2.4 mm/h, the larger of these events, would be a very successful outcome of any NFM intervention. This corresponds to a reduction of around 4.6 mm/h or 65%. It should be borne in mind the potential for large degree of uncertainty in the rating of these discharges, particularly at the extreme levels seen during Storm Desmond.





## 4.2    Identification and modelling of intervention areas

The catchment was divided into 8 response units according to the topographic wetness index (TWI, see Beven & Kirkby, 1979). JFLOW was run across the catchment using a design event of 30 year return period. Areas that accumulated significant water depths, such as natural depressions, flow pathways or small channels, were tagged as suitable candidates

for enhanced storage. Their areas were then constrained in size to between 100 and 5000 m² and those within 2m of roads and buildings excluded. In principle, times to peak at the outlets of individual subcatchments estimated from the modelled runoff  designed event could be used to exclude faster responding areas from the introduction of features designed to slow their flood waves and thus mitigate the possibility of synchronisation. This was not undertaken in the Eden, however, but Hankin et al. (2016) applied in the approach in the Kent headwater catchments to the SW. In addition, areas in peat with

slopes of greater than 6% were omitted as not conforming to current peat management practices (Moors for the Future Partnership, 2005). This yielded 4500 distinct sites of average area 506m², occupying 4.0% of the catchment, with a potential of just over 8 million m³ static storage.

The areas were tagged as being a unique HRU in the catchment discretisation, overriding any underlying classification determined from the TWI. This unit was treated as though a single aggregated feature bunded or dammed by a "leaky"

barrier 1m in height with upslope sides open to receive surface runoff. Overflow was directed over the top of the barrier in the same direction as the original distributions given by the surface flow weighting matrix. The specific overflow per unit length along the top side of the feature was calculated as though for a broad-crested weir of width 50cm. Discharge coefficients are taken from the appropriate entries in the tables provided by Brater and King (1976). The unit took the same hydrological parameters as the surrounding regions. Three scenarios were considered, labelled RAF1, RAF10 and RAF100 ,

corresponding to Λ factors equivalent to residence times of 1, 10 and 100 hours., respectively.

## 4.3    Monte Carlo analysis and identification of behavioural model realisations

An initial calibration exercise sampled 5000 parameters non-informative, uniform, prior distributions with ranges given in Table 1. The observables used to calculate likelihood weighting score were: $A_c$, the maximum saturated contributing area, or area proportion of the catchment that generates overland flow, the Nash-Sutcliffe statistic (NSE), and $q_{max}$, the maximum

simulated discharge relative to the observed rated value. The acceptability criteria for $A_c$ were derived from considerations of physically-feasible values obtained from field observations, such as those in Beven and Wood (1983) and Chappell et al. (2006), and subjective, expert opinion of the likely values in an extreme event as such Storm Desmond. The criteria observables are given in Table 2, alongside their limits of acceptability. The likelihood score for the NSE was the actual value calculated from the simulated discharges versus the observed, rated values. For the others, the likelihood score was

triangular in the corresponding acceptability intervals, with value of unity at the midpoint of the range. The overall weighting score for a realisation is then calculated by taking the mean of the individual scores. The behavioural sets identified by




applying the limits shown in Table 2 are then used as the basis for investigation of the effects of applying a number of NFM scenarios.

[INSERT Table 2]

## 5 Results

Of the 5000 realisations undertaken, 384 were identified with outputs that met the acceptability criteria given in Table 2. shows the discharges simulated by these behavioural realisations alongside the observed rated discharges at the EA gauge at Great Musgrave bridge.

The "dotty" plots in Figure 4 show the GLUE weightings for the three metrics used as the basis for selection of these acceptable cases. The maximum saturated area, $A_c$, shown in the leftmost plot, has a discontinuous shape. This is because

when a response unit reaches saturation it contributes its entire area at once. Within the behavioural realisations only a limited number of HRUs, along with the RAF unit, ever contribute saturated surface flow, leading to just seven distinct values for $A_c$. Points corresponding to likelihood weightings for realisations producing these distinct values of $A_c$ are shown in the leftmost plot using a different colour for each. The same colours are applied to points corresponding to the same realisations in the other plots. The stratified appearance of the NSE plot is a by-product of its correlation with the

contributing area; the correlation with the maximum predicted discharge is less clear. A value of $A_c$ around 65% is associated with the best NSE fits, but is spread throughout the maximum discharges. The bias towards higher values suggests that realisations producing more fast overland flow better reflect the storm response in this period. This would be consistent with the extreme nature of the storms and, albeit limited, observational evidence (e.g. Marsh et al., 2016).

[INSERT Figure 4]

Parameters for each of the 384 accepted cases were applied to catchment models modified to reflect insertion of RAF networks with each of the residence times considered, and the simulations re-run. The statistics for each of the events and intervention levels are given in Table 3 and the impacts on the arrival time of the main peaks of each in Table 4. The Kolmogorov-Smirnov test included is a non-parametric approach to comparing empirical distribution, equal to the maximum vertical separation of the CDF of the discharges. This allows an evaluation of the relative effectiveness of each intervention.

Reduction in peak $\Delta q$ for a RAF case is defined as the difference between a base line case and intervention simulation based on this case.

[INSERT Table 3]

Figure 5 shows, as rainfall equivalent specific storages across the entire catchment using the 90% percentile weighted realisations, for the aggregated RAF unit throughout the whole simulation period. The crest height will exceed the maximum

storage across the entire unit of at 1m as overflow across the top of the features, thus providing more storage than predicted by the hydrostatic analysis.

[INSERT Figure 5]





In the RAF1 case the areas drain quickly and appear to never fill completely. The available storage therefore appears not to be utilised effectively: the maximum utilised, at the peak of Storm Desmond, is 29% of the theoretical hydrostatic capacity, equivalent to 2,446,203m³ volume of storage retained across the catchment. The corresponding effect on the hydrograph is small, with a mean reduction in the peak of Storm Desmond of 4.3%; a few cases even show a small increase. For RAF10

the features appear under-utilised in the earlier storms, peaking at about 50% capacity, and recover almost all their capacity in the recession period after Storm Barney. Due to hydrodynamic storage utilisation exceeds 100% at the peak of Storm Desmond, with maximum filled storage volume across the catchment of just over 10 million m³.The impact of the additional storage is significant in the final storm and reduces the peak by an average of 14.4%, the greatest impact of any of the interventions. Combined with more conventional FRM measures this could have significantly mitigated the effects even of

this extreme event.

In the RAF100 case the features fill completely during the course of Storm Abigail near the start of the period and the drain-down is too slow to allow complete recovery of capacity before the final event. They are overflowing during much of the event, with the excess following the fast pathways to the channel blocked by the RAFs The impact is therefore much reduced compared with the 10 hr case, with a mean reduction of 4.8% in the peak.

Ensemble hydrographs through each of the named storms are shown in the following figures. These present the discharges simulated for the unmodified catchment model using the parameters for each of the acceptable cases against those produced by applying the corresponding parameters to catchment models with the insertion of RAFs.

**[INSERT Figure 6]**

In the initial storm, Abigail, the first peak is attenuated as much by the RAF100 case as the RAF10 case, but in the second

peak RAF10 again provides much greater reduction. The RAF1 case has the lowest impact on all of the peaks. The arrival of the main peak is retarded most by the RAF100 case, with a median delay of 30 minutes. The later peak appears to be brought forward marginally by the RAF100 cases, however. This may indicate that the RAF unit has run out of storage and is delivering water downslope  by overflow. In all cases the recession curve is extended by the intervention, indicating that stored water is draining for some time after the storm peak.

**[INSERT Figure 7]**

In the second event the RAF100 case appears to have most impact than any of the other cases across the initial peak but then becomes less effective than RAF10 through the main peak. There is less delay to the arrival of the main peak than for Abigail, with a median delay of only 15 minutes for the RAF10 and RAF100 cases. In the RAF10 case the median brings the peak forward by 15 minutes. This might indicate that their effect has been to slow down fast-responding catchments so that

their flood waves contribute more to the rising limb of the overall storm hydrograph.

**[INSERT Figure 8]**

In this largest event the RAF10 case significantly outperforms the others, suggesting that it has recovered much more capacity. There is virtually no attenuation from the other cases and the peak is hardly delayed for RAF1 and RAF100**.** From Figure 5 it is clear that the RAF100 units fill quickly during Abigail and are full for the duration of Barney and Desmond,



with only a week-long period in later November when they recover a little capacity through drainage during the recession period of Barney.

The RAF10 case has the greatest impact on the flood peak for all storms, with its advantage over the other cases increasing through the period. In Abigail the K-S distance of the RAF10 case is 150% that of the RAF100 case, whereas across Desmond this has increased to 350%.

## 6  Discussion

The surface routing algorithm is computationally very efficient, particularly as it can be solved analytically. The storage-based approach to RAF representation is straightforward and allows examination of relevant characteristics such as the drainage times and multiple model runs that apply different network configurations. More sophisticated physically-based modelling of surface flow will introduce computational overheads without accounting for large uncertainties in the input, and may not add greater insight into the catchment response.

The RAF10 case was significantly more effective across the largest event than the other two cases, as features had recovered capacity during the previous recession and drained sufficiently to have an impact on the flood peak in Storm Desmond. The mitigation was less across the earlier events but, given they did not cause significant flooding, this was not a consideration. This behaviour, whereby features that fill less quickly during intermediate events but also recover capacity that can mitigate later, more damaging, flood peaks, was simulated for in-channel NFM measures by Metcalfe et al. (2017).

NFM practitioners have previously designed schemes on the basis of the additional storage (hillslope and channel) that, when fully utilised, would retain sufficient runoff from a particular magnitude of event to reduce its peak to below the level where flooding would occur (see for example Nicholson et al., 2012). These results, however, provide further evidence that the theoretical storage provided by an NFM scheme can be significantly underutilised, assuming it is configured so that it can recover capacity effectively between storms events. It has already been suggested that the effectiveness in reducing flood risk of a NFM intervention is not a simple function of the additional storage it provides, but also of its contribution to desynchronising a subcatchment's flood wave with those downstream (Thomas and Nisbet, 2007; Blanc et al., 2012).

A more sophisticated approach to implementing NFM is required. For instance, the locations on the hillslopes of interventions (such as RAFs) are likely to be a significant factor on a scheme's performance as they will influence flood wave timing. In this study the network width approach to routing channel flow was applied across the entire headwater catchment, and flood wave size and timings were not available for the individual subcatchments. It was thus unclear from the results what proportion of attenuation of storm hydrographs were due to desynchronisation of flood waves, as opposed to simple retention of surface run-off. A regression analysis similar to that undertaken by Pattinson et al. (2013), including as predictor variable the hillslope storage provided by the RAFs, could provide insight on the relative contributions of additional hillslope storage and flood wave desynchronisation on downstream flood mitigation. This can be done in a further analysis, but does raise the question of how to calibrate meaningfully flood wave velocities for the individual subcatchments.





Metcalfe et al. (2015) showed that the runoff predictions simulated by Dynamic TOPMODEL stabilised at around eight to twelve subdivisions of the catchment. In this study eight HRUs were used: despite being at the lower limit of the region of stability identified by Metcalfe et al. (2015), the model gave good fits to the observed hydrographs, with many realisations exceeded efficiencies of 0.9 (although the uncertainties introduced through rating of discharges at extreme storm levels

should be noted). The values of the maximum saturated area, $A_c$ were highly discontinuous, however, as when the response units start to produce saturated flow they contribute their entire area to the metric. A more fine-grained discretisation would produce a more continuous distribution which might be advantageous in terms of better identification of saturated contributing areas, in particular how the positioning of RAFs in proposed schemes may influence runoff from these areas.

The features appeared to have an effect through the course of even the largest storm, albeit that the slowest-draining cases

were full and overflowing for much of the simulation period. There is, however, a possibility that features will be unable to withstand hydraulic loading across such extreme events, or across a series of storms. Complete or partial failure could lead to debris being introduced to the flood waters and potential damage or blockage of downstream infrastructure which will increase risk of flood damage. Ideally, these scenarios should be incorporated in a risk reduction – cost matrix but there is, as yet, little information about the potential for failures that can be used to estimate residual risk., although initial findings

suggest that failure sequences can behave non-linearly.

The RAFs were treated as a single unit in the catchment discretisation. A more considered approach to grouping features would be to categorise them by their type (e.g. bunded or screened ephemeral channels, enhanced hillslope depressions, or leaky dams), their geometries and their position on the hillslopes. The number of possibilities will increase rapidly according to the number of varieties and configurations defined, but these will be constrained by considerations of realistic design and

implementation issues. In addition, the simplified representation used in this study allows for relatively rapid investigation of many different configurations. This approach can be incorporated into a sensitivity analysis to determine which characteristics are most important to the impact of RAFs on the storm response. A Monte Carlo approach selecting from multiple design event sets, for example generated by the method of Keef et al. (2013), could help assess the robustness of the conclusions drawn from modelling and help to site features more strategically. It could also help identify situations where

flood waves become synchronised by emplacement of features at different catchment scales.

## 7   Conclusions

This study has analysed the performance across a series of extreme events of a natural flood management scheme incorporating many "leaky" hillslope run-off attenuation features within the headwaters of a large catchment. The model incorporated a simplified overland routing module and achieved a high level of efficiency in simulating the observed

discharges. The best base line simulations were applied to the catchment model with NFM features incorporated into the model. RAFs were simulated using a simple aggregated linear store model. It has showed that their aggregated impact could have significantly attenuated the flood peak, by up to a maximum of almost 30%, even during the largest storm, but that their





impact was contingent on the permeability of the RAF features that allowed them to recover capacity between events. The study demonstrates that uncertainty estimation can be applied to NFM in this context. A well-established uncertainty analysis framework was used, whereby multiple realisation of a hillslope runoff model applied to the events were enacted and scored against physically measurable characteristics and selected according to limits of acceptability of those observables. This allowed results to presented to project and catchment stakeholders alongside meaningful estimates of their uncertainty.

The RAF representation is efficient and allows investigation of many different configurations whilst retaining the important aspects of their behaviour and impacts, namely storage addition and residence times It contains many assumptions and simplifications, however, and a key aim of further work will be to determine whether other significant characteristics are adequately simulated in this representation, and which will require refinement. For example, in actual applications there is likely to be a more complex storage-water depth relationship than the straightforward equivalence used here. Hydrodynamic analysis of simulated individual structures sited in realistic topographic representations may provide insights into the applicability of the simplification across a range of loading scenarios.

The RAF features considered in this study were lumped into a single HRU to maximise computational efficiency. A more sophisticated approach could utilise many more feature classifications derived from position on the hillslope, distance from access tracks, the channel and sources of construction material, type of location (e.g. within ephemeral channels, shallow hillslope accumulation areas or on the floodplain). More work will be needed to determine which of these characteristics will be most significant and how well classifications reflect actual implementations by NFM practitioners.

It may be that the most beneficial effects of RAF emplacement are likely to be seen on a smaller scale, in reaches immediately downstream of a feature or sets of features. It could be, even given the overall mitigation effect, that some asynchronous flood peaks in the unmodified catchment model became synchronised when the RAFs were introduced and thus reduced the overall effectiveness of the interventions. Much further work is required to better understand the synchronisation problem, particularly as catchment scale increases.

Obtaining observational evidence to support modelling predictions will be difficult in the field as, by their nature, the extreme events that load features are rare and gradual processes such as sediment deposition difficult to measure or simulate everywhere. An innovative experimental approach is needed to address these questions. Detailed hydrometric data are required, collected by instruments such as stage and flow gauges upslope, downslope and within features. However only 6% of NFM schemes in the UK currently have any type of monitoring (JBA Trust, 2016). The effects of the RAFs on the overall flood hydrograph will be increasingly difficult to discern as the catchment size increases, including the potential for synchronicity effects. The methodology used in this paper, however, extended to incorporate better-supported representations of small scale impacts due to feature emplacement, may provide a productive way to take forward research into natural flood management and its effectiveness.





## Acknowledgements

This work was funded by the JBA Trust (project number W12-1866) and managed by the Centre for Global Eco-innovation (CGE project number 132)



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



**Tables**

Table 1: Runoff model parameter and ranges applied in calibration and uncertainty analysis.

| Parameter | Description | Units | Lower | Upper |
|---|---|---|---|---|
| $v_{of}$ | Overland flow velocity | m/h | 1 | 150 |
| $m$ | Form of exponential decline in conductivity | m | 0.0011 | 0.033 |
| $srz_{max}$ | Max root zone storage | m | 0.1 | 0.3 |
| $srz_0$ | Initial root zone storage | % | 20 | 100 |
| $v_{chan}$ | Channel routing velocity | m/h | 500 | 5000 |
| $ln(T_0)$ | Lateral saturated transmissivity | m²/h | 3 | 12 |
| $sd_{max}$ | Maximum effective deficit of saturated zone | m | 0.5 | - |
| $ex_{min}$ | Minimum surface storage | m | 0 | - |
| $ex_{max}$ | Maximum surface storage | m | 1 | 1 |

Table 2: Observables collected for each model realization and acceptability criteria applied

| Metric | Description | Units | Criteria applied |
|---|---|---|---|
| NSE | Nash Sutcliffe statistic | - | >= 0.85 |
| $A_c$ | Maximum saturated contributing area | % | [10, 95] |
| $q_{max}$ | Specific discharge at largest storm peak | mm/h | [5.2, 8.2] |

Table 3. Statistics for each RAF interventions across the named storms in the simulation period: $\Delta q_{max}$ = maximum relative reduction in peak (%); $\overline{\Delta q}$ = mean relative reduction in peak (%); K-S = Kolmogorov-Smirnov Statistic

| | Abigail | | | Barney | | | Desmond | | |
|---|---|---|---|---|---|---|---|---|---|
| Name | $\Delta q_{max}$ | $\overline{\Delta q}$ | K-S | $\Delta q_{qmax}$ | $\overline{\Delta q}$ | K.S. | $\Delta q_{max}$ | $\overline{\Delta q}$ | K.S. |
| RAF1 | 13.8 | 6.4 | 0.14 | 22.2 | 9.4 | 0.12 | 17.7 | 4.3 | 0.45 |
| RAF10 | 49.6 | 30.2 | 0.77 | 66.4 | 26.8 | 0.43 | 25.4 | 14.5 | 0.82 |
| RAF100 | 42.5 | 16.6 | 0.33 | 64 | 20.4 | 0.30 | 28.7 | 4.9 | 0.44 |





**Table 4. Mean and median delays (h) to simulated peaks of each of the named storms for the RAF intervention cases compared to the corresponding unmodified cases.**

|  | Abigail | | Barney | | Desmond | |
|---|---|---|---|---|---|---|
| Name | Mean | Median | Mean | Median | Mean | Median |
| RAF1 | 0.33 | 0.25 | 0.35 | 0.25 | 1.6 | 0 |
| RAF10 | -0.13 | 0 | -0.16 | -0.25 | 6.18 | 6.5 |
| RAF100 | 0.48 | 0.5 | 0.33 | 0 | 1.59 | 0 |





**Figures**

**Figure 1: Hydrodynamic accumulation areas within Eden identified by JFLOW analysis for a designed storm of return period of 30 years (Hankin et al., 2017). Maximum water depths are indicated, and areas that exceed the threshold depth and other criteria (minimum area, slope angle and proximity to roads and buildings) are highlighted as potential sites for RAFs**





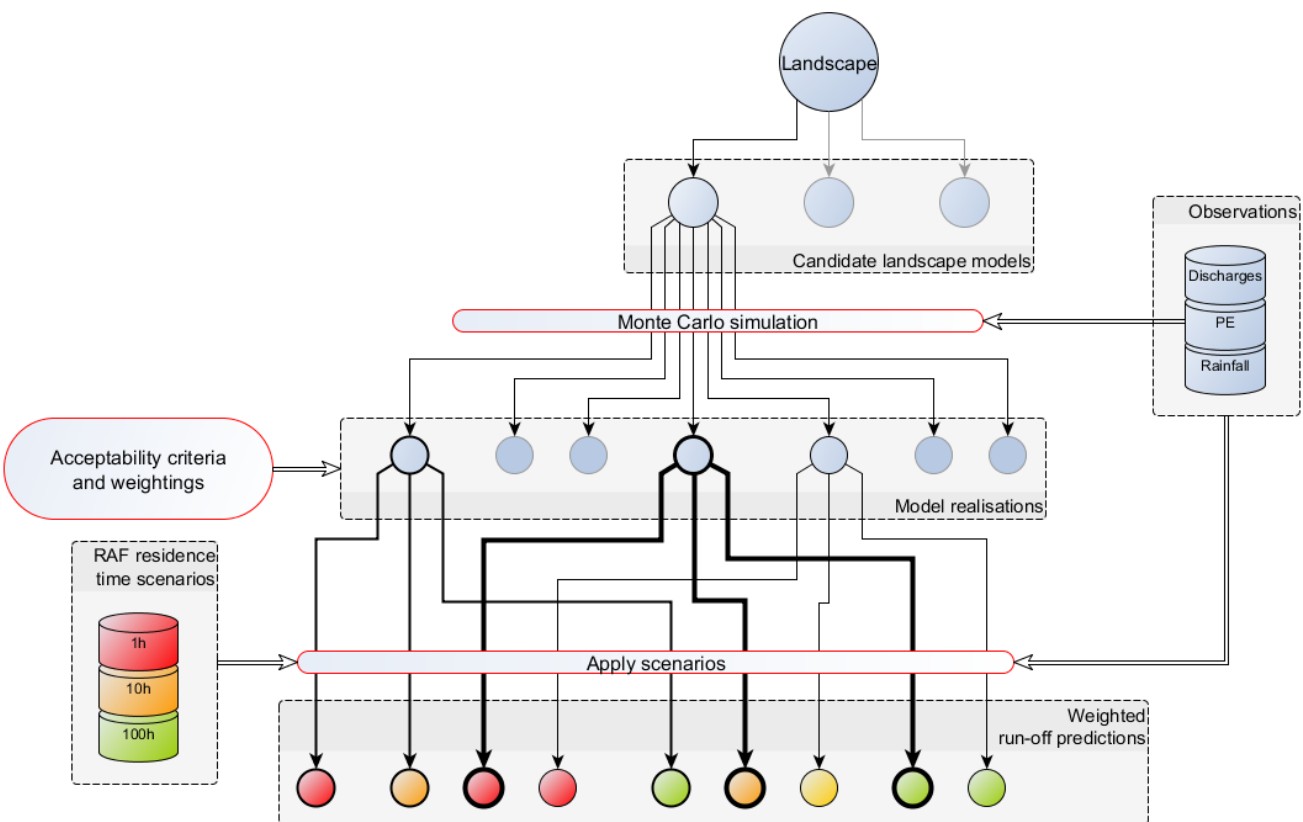

**Figure 2: Suggested work flow diagram for Monte Carlo simulation of storm runoff, and selection and weighting of behavioural realisations and application of NFM scenarios for forward prediction of change. The weight of lines leading from acceptable simulations reflects the weighting likelihood score in the validity of that realisation.**



**Figure 3. Study catchment, the Eden headwaters to Great Musgrave Bridge (223km²), showing context within Cumbria, UK, predominant land cover types and location of TBR rain gauges and gauging stations and predominant land cover. Woodlands for Water opportunity areas are shown. These were applied in another application of the NFM modelling framework developed for the project described, which not discussed in detail here.**





**Figure 4. Simulated discharges at Great Musgrave Bridge across the calibration period described in the main text for behavioural realisations alongside rated observed discharges. The three named storms are indicated. Rainfall is interpolated between the gauges shown in Figure 3.**




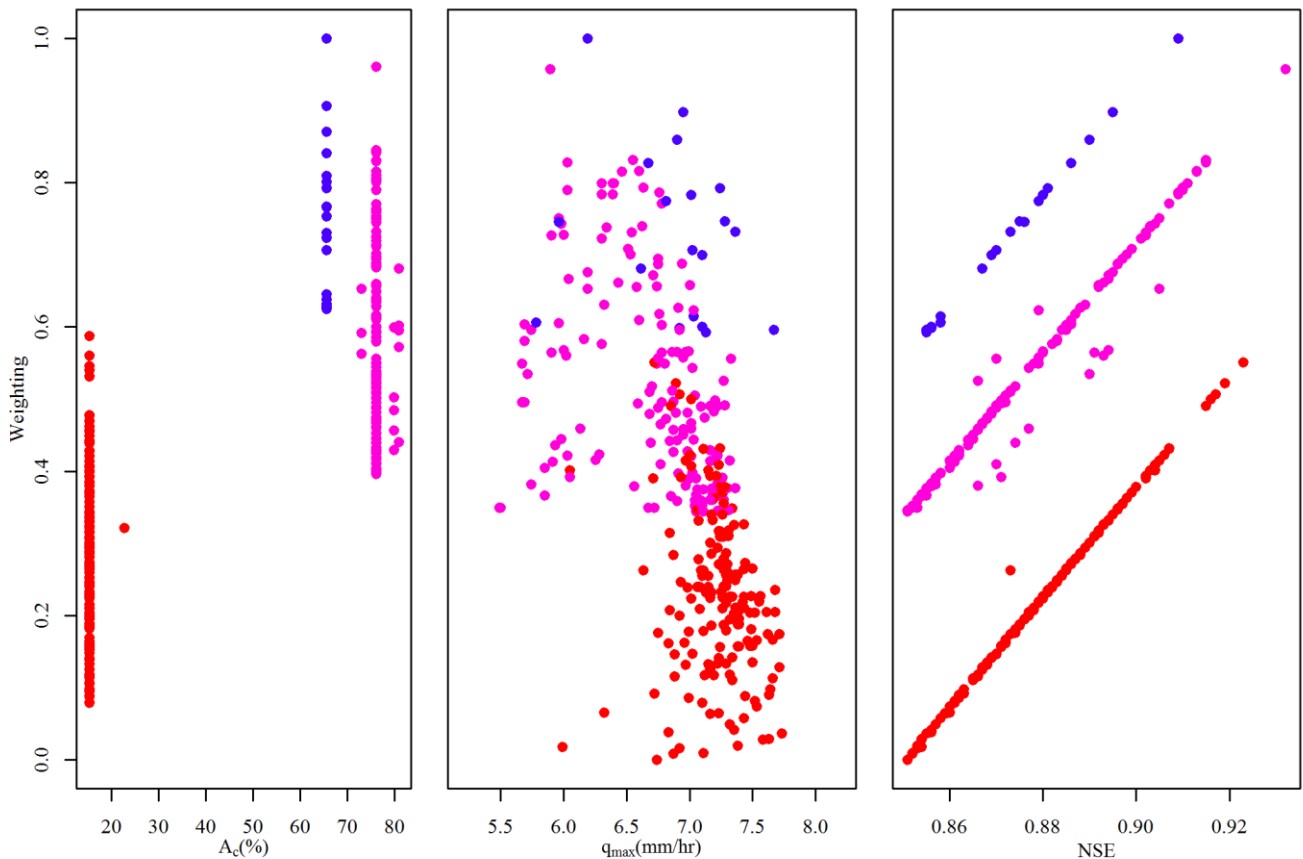

**Figure 5. GLUE "dotty" plots showing overall weighting (likelihood) scores for each of the 348 behavioural runoff simulations identified against the three model outputs described in the text. The discontinuous appearance of the maximum saturated contributing area $A_c$ is due to the relatively coarse discretisation applied such that once a HRU begins to produce any saturated overland flow, its entire area is added. Each unique $A_c$ value takes a separate colour that is carried through to corresponding points in the other plots.**



**Figure 6. Surface excess storages, expressed as specific rainfall equivalent, across one of the lumped RAF units with maximum storage 1m through a single intervention cases and for the three mean residences times considered. The slight excess at the peaks of the storm reflects the weir crest height of the overflow function applied.**





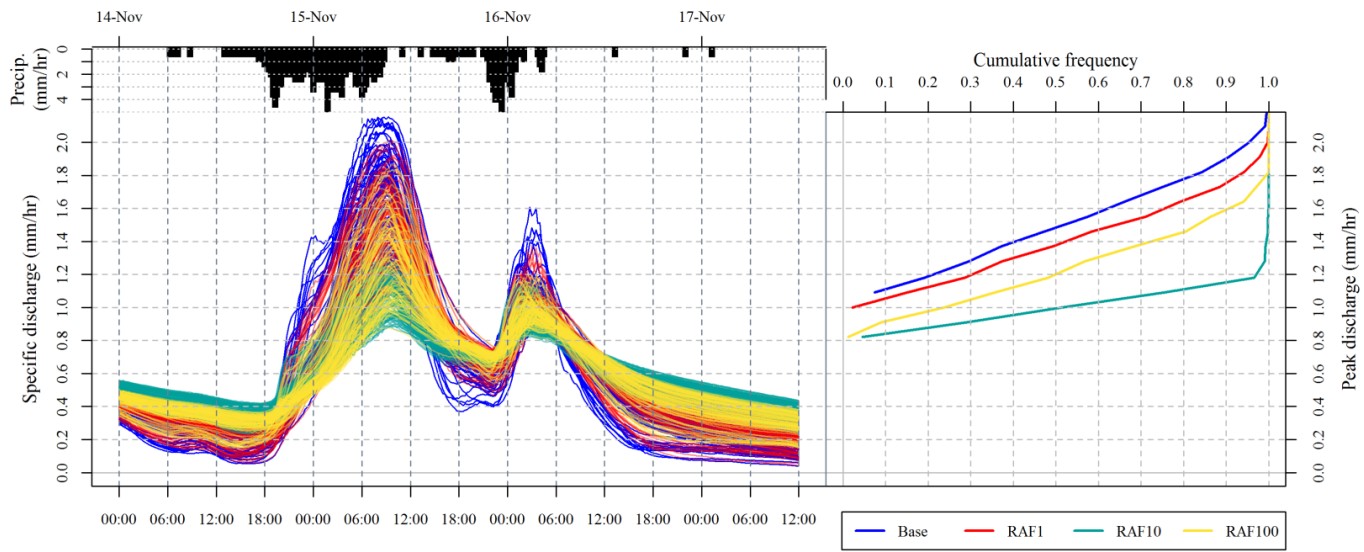

**Figure 7. (L) 90 percentile weighted scored baseline and corresponding RAF intervention cases through Storm Abigail. (R) Cumulative frequency plot peak discharges for base and intervention cases. The K-S statistic for each is the maximum horizontal displacement between their lines and the leftmost, unaltered cases. Note that, in order to share the same vertical axis, the Cumulative frequency plot is transposed relative to convention.**





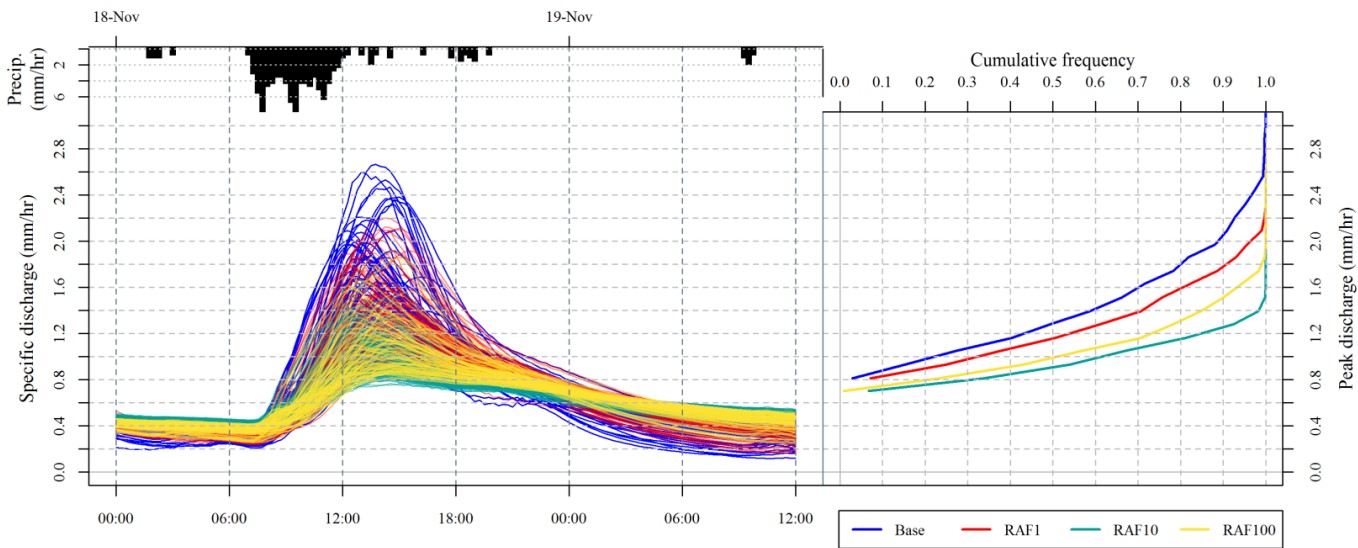

**Figure 8. (L) Selection of base line and corresponding RAF intervention cases through Storm Abigail. (R) Cumulative frequency plot of peak discharges for base and intervention cases.**




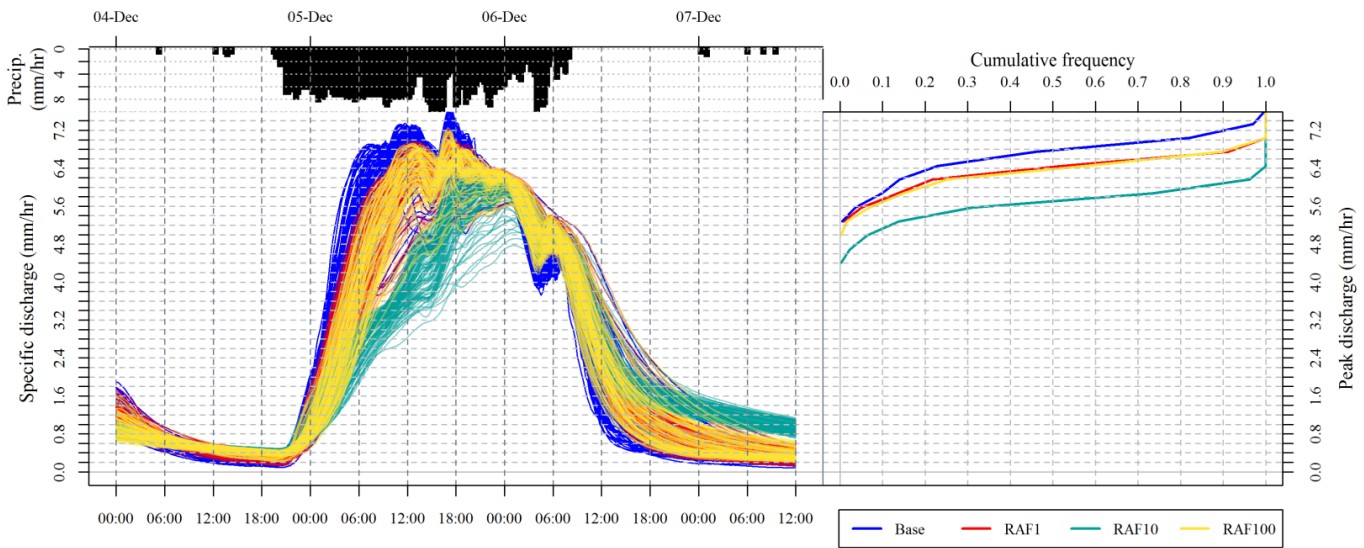

**Figure 9. (L) Selection of base line and corresponding RAF intervention cases through Storm Desmond. (R) Cumulative frequency plot of peak discharges for base and intervention cases.**