# Peer review of "Simplified representation of runoff attenuation features within analysis of the hydrological performance of a natural flood management scheme"

_Hydrology and Earth System Sciences, 2017_

## Referee Comment (RC1) · S. Dadson (Referee) · 4 Sep 2017

This paper investigates the extent to which natural flood risk management might have mitigated the effects of a series of extreme storms in northern Britain. The paper represents an important contribution to the literature which will be of broad interest to hydrologists concerned with the theoretical aspects of this problem and to applied workers involved with flood risk management and I recommend publication in HESS subject to a number of revisions to clarify the text and to resolve a series of queries outlined below.

1. The term "runoff attenuation features" is establishing itself in the literature but it is an

[Figure]

unfortunate use of overcomplicated language where simple language will do. In some cases RAFs are simply what most people would call ponds! In other cases the use of the term obscures the relative effects of storage and attenuation and glosses over the variety of interventions that might be possible by lumping them together as RAFs. My suggestion is to be clear about which types of intervention are being referred to at the outset, and to prefer simple language if possible.

2. Some clarity on how the drainage time constant is estimated (p3. Line 5) is essential. It appears to be allowed to range over two orders of magnitude for the purposes of the uncertainty estimation exercise. Fine if that's all that can be done but some discussion of what values the parameter might sensibly take is warranted in a paper of this kind. Is it considered an observable parameter that an engineer might design to or control?

3. Given the simplicity of the model there is an overly confident equation of model performance with the operation of the "real world". For example on p. 3 line 20 after only just having introduced the modelling approach it is declared that the model "can be used to examine the drain down, filling and possible overflow of these features during the course of storm events." I understand the point that's being made but it would be as well to note that the model can only provide an understanding commensurate with the fidelity of its representation of the interventions. The opportunity on p5 line 20 to describe whether different types of feature behave as modelled is also missed (and also on p6 line 25 when it is stated that tree shelterbelts can be modelled using this approach too). I suggest that some additional text in these parts of the manuscript would help to improve the paper.

4. A key part of the research problem is to distinguish between storage effects and network / wave propagation effects. This is very clearly articulated on p4 lines 21ff, an explanation that might more helpfully be given earlier in the paper. The comment p17 line 22 that network effects are important is appropriate to the discussion (and has been raised in the literature before) but it is hard to see what new evidence is offered for this view in the present manuscript.

5. The tables indicate the results of the study very clearly but the text is a little more equivocal, and is unclear in parts. For example, it would appear from the tables that the RAF10 intervention shows a 30% reduction in peak flow – an important effect - but this is not discussed until the conclusions and is only mentioned in passing (p16 line 32). More detailed presentation of the results in Section 5 is warranted, in my view.

Minor comments

There are a number of typos / grammatical mistakes / errors of punctuation throughout, which might usefully be corrected at this stage.

The title of the paper somewhat understates the conclusions of the piece. In its current form the title suggests that the paper offers only an improvement in method. No doubt an improvement in the simulation of runoff attenuation features would be worthwhile in its own right but the work presented takes the ideas further and in fact evaluates and seeks to draw conclusions on the performance of systems of RAFs under multiple storm conditions. I'd suggest that the title be edited to reflect the wider scope of the paper as written.

Please clarify the statement at p14 line 6: "Due to hydrodynamic storage utilisation exceeds 100% at the peak of Storm Desmond..."

Table 4. Consider the implied precision of the modelled estimates when quoting simulated peak delays in hours to two decimal places.

Simon Dadson simon.dadson@ouce.ox.ac.uk

---

## Referee Comment (RC2) · Anonymous Referee #2 · 7 Sep 2017

The manuscript presents a simplified modelling approach to evaluate the effectivity of spatially distributed runoff attenuation features on a sub-regional scale during rainstorms. The topic and the presented methodology are definitely very interesting for both scientific community and landscape managers. The presented study thematically fits well for publication in HESS journal.

My comments:

I must admit I have some difficulties to follow the text in the first chapters (p. 2-12). Structure of the manuscript is a little bit confusing; it does not follow the classical IM-RAD approach. Same information are repeated on several places, paragraphs are

sometimes too long containing too many topics. Often I am not sure if the authors describe actual research or results from the previous studies (referred as Metcalfe at al., or Hankin et al.). Simplification of the text structure and rephrasing of long sentences would very much help to increase the readability and overall appeal. Concrete examples follow:

C#1: p.3, l. 9: I don't clearly understand which PROJECT you refer to. Is it related to Hankin et al. (2017) or another previous work? Or do you address actual research presented in this manuscript? Similarly, on the rest of p. 3, where several times "objectives of the project" are mentioned. At the moment the text reminds me more of a project proposal than an article. My suggestion is to move most of the chapter 1.1 directly to the Introduction. In Aims and Objectives simply state the "aims, objectives, hypotheses" (even the bullets will make it). I would also omit the hints of the used methods in Introduction (eg. different levels of leakiness through the walls, Dynamic TOPMODEL, Storm Desmond etc.) and leave it for the following chapters. Just to make the text easier to follow.

C#2: p. 4, ch 1.2 – I suggest to move the whole chapter before Aims and Objectives. This is still pure introduction.

C#3: p. 7, 1st paragraph: it was already stated before that distributed modelling is computationally demanding. You have already introduced the simplified approach that you use. This paragraph should be a part of Introduction.

C#4: p. 12 I don't understand how 8 HRU units correspond to the JFLOW simulation. Was JFLOW used on 8 separate subregions? Or are the HRUs related to the TOPMODEL simulation? Please, clarify. What is approximate size of the HRUs (are they similar in size)? Can you include HRUs boundaries on Fig. 3?

C#5: Figures need to be improved. The lines/dots have similar colours (eg. fig 5b – orange, pink). Yellow lines are not visible (fig. 6). Corresponding plots (fig. 4 &6) have different starting and ending dates, precipitation is not consistent. Etc.

C#6: I don't understand to the description of the GLUE results (p.13, l. 13). Why the Ac around 65% gives the best NSE? It does not seem like that on Fig. 5.

C#7: in abstract and introduction you stress out that the unfortunate synchronization of the RAFs overtopping may cause serious problems downstream, therefore the RAFs network must be designed wisely. I do not see how you solved the desynchronization problem in the Results (slightly mentioned in Discussion). Can you please address this issue in more details in Results, Discussion and Conclusion?

Minor and technical comments:

There are many typos and grammatical mistakes in the text, units often follow the number without added space in between (eg. p.4, l. 13-14: 27000m3, 9%). Cited authors in the text sometimes do not agree with the list of references - check the years and Author's spelling please (eg. p. 4, l. 20: Pattison or Pattinson, l. 13: Ghmire, p. 10 l. 9: Binley & Beven should be Beven & Binley?, etc.). Sometimes decimal separator for thousands is used (p.4, l. 17), mostly not.

p.1 Abstract: The abstract is very well written, I enjoyed reading it! Only the last sentence is misleading. At the moment "ways in which features could be grouped more strategically" are not given in the manuscript (there are general hints which are not supported by the simulation results). This sentence should be excluded or (preferably) the information included in the manuscript.

p.2, l. 25: If JFLOW analysis and the workshop were used for this study, the information should be moved to Methods, rather than to state it in Introduction. The workshop and it's outcomes are not further mentioned.

p.2, l. 30: The sentence repeats the information from the previous sentence (as RAF is also flood mitigation measure).

P3, l. 11: word repetition (would)

p.4, l. 14: word repetition (in)

p. 5, l. 22: Can you please specify why subsurface routing allows more flexible HRU aggregation in this case? How is it related to this study where the surface runoff is of the primary interest? Is this information related to the following sentence "Of particular relevance. . ."?

p.7, l. 29: Which "study case"? The case study area has not been introduced yet. The chapter 2.3 mixes introduction and methods together.

p. 8: Can you justify or discuss how multiple RAFs are lumped into single RU? I don't think that the weir equations are necessary to be presented here.

p.11, Study site: describe the instrumentation that provides the validation data, please. How the discharge is monitored (what kind of flume/weir, what capacity)? P.11, l 22. – the other gauge should be already mentioned together with the GM Bridge gauge.

p. 12, l. 20: can you also include the lambda factor values?

p. 13, l. 10: 7 distinct values of Ac are mentioned in the text, only 3 are on Fig. 5.

p. 13, l. 18: This paragraph belongs to Methods

Some literature is missing in the references: Daddson et al. (2017), Beven & Blazkova (2004), EA (2009), Marsh (2016), Beven & Wood (1983), Chapel et al. (2006)

Fig. 3: I suggest to include the HRUs boundaries. Position of the NFMs would be also interesting, but there are maybe to many of them.

Fig. 4: Why are there more red lines? I would think that there is only one observation in the single gauge. The Precipitation bars are too thick. In the given scale the daily precipitation amount on 13th Nov would be app 240 mm.

Fig. 5: What are the thin lines with max excess storage around 0.2 m, visible after Desmond storm? Why are they so different from most of the simulation results (for all the RAFs)?

---

## Author Comment (AC1) · 9 Oct 2017

**Comment #1**

The term "runoff attenuation features" is establishing itself in the literature but it is an C1 HESSD Interactive comment unfortunate use of over-complicated language where simple language will do. In some cases RAFs are simply what most people would call ponds! In other cases the use of the term obscures the relative effects of storage and attenuation and glosses over the variety of interventions that might be possible by lumping them together as RAFs. My suggestion is to be clear about which types of intervention are being referred to at the outset, and to prefer simple language if

possible.

**Response #1**

We agree with the Reviewer that the lumped term can be misleading or inaccurate and could be considered unnecessarily technical. Our suggestion is to use "enhanced hillslope storage", and to refer to ponds, gully blocking etc where necessary to clarify what this term is referring to.

**Comment #2**

Some clarity on how the drainage time constant is estimated (p3. Line 5) is essential. It appears to be allowed to range over two orders of magnitude for the purposes of the uncertainty estimation exercise. Fine if that's all that can be done but some discussion of what values the parameter might sensibly take is warranted in a paper of this kind. Is it considered an observable parameter that an engineer might design to or control?

**Response #2**

Given the assumption of linear storage – discharge relationship (p6, Line 14), residence time is simply the reciprocal of the constant of proportionality in the relationship. We chose the large range to ensure that both very fast draining and very slow draining cases were considered. A much larger range would have been examined but this could have obscured the conclusions drawn. The linear store assumption is not necessary for the solution of the system of equations given on p.6 Line 19, although an analytical solution would no longer be possible. Any storage-discharge can be specified for individual response units, and the system solved numerically. This will be clarified, and it will be stressed that the example relationship is the simplest that could be used and is likely to be more complex. With the applied relationship, estimation of the drainage constant is an issue that will need to be addressed in further experimental work. Applying a more sophisticated hydraulic model, for example to leaky gully barriers , could be more realistic and, in theory at least, related directly to the configuration of a feature in

terms of size of and spacing between members etc. This however would introduce further parameters but the identification of feasible ranges could be the subject of further work.

**Comment #3**

Given the simplicity of the model there is an overly confident equation of model performance with the operation of the "real world". For example on p. 3 line 20 after only just having introduced the modelling approach it is declared that the model "can be used to examine the drain down, filling and possible overflow of these features during the course of storm events." I understand the point that's being made but it would be as well to note that the model can only provide an understanding commensurate with the fidelity of its representation of the interventions.

**Response #3**

This is true and we will attempt to qualify these statements where they occur.

**Comment 4**

The opportunity on p5 line 20 to describe whether different types of feature behave as modelled is also missed (and also on p6 line 25 when it is stated that tree shelterbelts can be modelled using this approach too). I suggest that some additional text in these parts of the manuscript would help to improve the paper.

**Response #4**

We agree with the first point and we shall try to expand on this section. The example of tree shelterbelts is not actually given as a case of an intervention that could be modelled in this approach, so this section will be clarified or removed. It is included as example of how spare downslope soil capacity could be introduced by an NFM-type intervention, and thus that reinfiltration should therefore be considered and this can be handled by the model.

**Comment #5**

A key part of the research problem is to distinguish between storage effects and network / wave propagation effects. This is very clearly articulated on p4 lines 21ff, an explanation that might more helpfully be given earlier in the paper. The comment p17 line 22 that network effects are important is appropriate to the discussion (and has been raised in the literature before) but it is hard to see what new evidence is offered for this view in the present manuscript.

**Response #5**

We agree that this is a potentially very significant component of the effectiveness, or otherwise, of NFM schemes. Although the paper in itself provides no major evidence, it does, however, give a potential use of the model presented and will be the subject of further investigation. We shall qualify the statement with the caveat that the paper does not actually address the network synchronisation issue but the model presented does allow the possibility for this to be investigated.

**Comment #6**

The tables indicate the results of the study very clearly but the text is a little more equivocal, and is unclear in parts. For example, it would appear from the tables that the RAF10 intervention shows a 30

**Response #6**

Some additional text on what these table show is warranted, and, possibly, a change in what is shown. There is an extremely wide range of potential impacts, with magnitudes up to that quoted, but the majority are smaller. The impact on the first storm (Abigail) is largest as all schemes have their maximum storage. A better metric than the maximum difference would probably be the median value. Each realisation is associated with a likelihood determined as described in section 3 and 4.3, and the largest reductions in peak flow are generally associated with the least likely realisation. This could be
used to likelihood weight the values and achieve a figure for the mean reduction more representative of the corresponding interventions' actual overall impact.

**Comment #7**

There are a number of typos / grammatical mistakes / errors of punctuation throughout, which might usefully be corrected at this stage.

**Response #7**

We are grateful for the reviewer pointing this out and we will endeavour to eliminate these errors.

**Comment #8**

The title of the paper somewhat understates the conclusions of the piece. In its current form the title suggests that the paper offers only an improvement in method. No doubt an improvement in the simulation of runoff attenuation features would be worthwhile in its own right but the work presented takes the ideas further and in fact evaluates and seeks to draw conclusions on the performance of systems of RAFs under multiple storm conditions. I'd suggest that the title be edited to reflect the wider scope of the paper as written.

**Response #8**

Our revised suggestion is "A new method, with application, for analysis of the impacts on storm runoff and flood risk of widely-distributed enhanced hillslope storage".

**Comment #9**

Please clarify the statement at p14 line 6: "Due to hydrodynamic storage utilisation exceeds 100% at the peak of Storm Desmond. . ."

**Response #9**

The weir equation applied to model overflow out of the features will result in a small

head clear of the rim that will introduce more storage over and above the hydrostatic amount. This storage cannot be identified beforehand as its is not known how large this head will be. Hence, the static storage of the feature at "brim-full" is equated with 100% utilisation.

**Comment #10**

Table 4. Consider the implied precision of the modelled estimates when quoting simulated peak delays in hours to two decimal places.

**Response #10** We shall round these to 1 d.p.

---

## Author Comment (AC2) · 9 Oct 2017

**Comment #1**

I must admit I have some difficulties to follow the text in the first chapters (p. 2-12). Structure of the manuscript is a little bit confusing; it does not follow the classical IM-RAD approach. Same information are repeated on several places, paragraphs are sometimes too long containing too many topics. Often I am not sure if the authors describe actual research or results from the previous studies (referred as Metcalfe at al., or Hankin et al.). Simplification of the text structure and rephrasing of long sentences would very much help to increase the readability and overall appeal.

[Figure]

**Response #1**

These are extremely useful suggestions and we will revise the introductory sections accordingly.

**Comment #2**

p.3, l. 9: I don't clearly understand which PROJECT you refer to. Is it related to Hankin et al. (2017) or another previous work? Or do you address actual research presented in this manuscript? Similarly, on the rest of p. 3, where several times "objectives of the project" are mentioned. At the moment the text reminds me more of a project proposal than an article. My suggestion is to move most of the chapter 1.1 directly to the Introduction. In Aims and Objectives simply state the "aims, objectives, hypotheses" (even the bullets will make it). I would also omit the hints of the used methods in Introduction (eg. different levels of leakiness through the walls, Dynamic TOPMODEL, Storm Desmond etc.) and leave it for the following chapters. Just to make the text easier to follow.

**Response #2**

These suggestions are very helpful in terms of improving the introduction and will be included. The project was the Rivers Trust IP project undertaken August - November 2016, whereas the initial study (Hankin et al., 2016) was the winning entry to the DEFRA flood modelling competition. We will attempt to clarify.

**Comment #3**

p. 4, ch 1.2 - I suggest to move the whole chapter before Aims and Objectives. This is still pure introduction.

**Response #3**

We shall do this.

**Comment #4**

[Figure]

p. 7, 1st paragraph: it was already stated before that distributed modelling is computationally demanding. You have already introduced the simplified approach that you use. This paragraph should be a part of Introduction.

**Response #4**

As above.

**Comment #5**

p. 12 I don't understand how 8 HRU units correspond to the JFLOW simulation. Was JFLOW used on 8 separate subregions? Or are the HRUs related to the TOPMODEL simulation? Please, clarify. What is approximate size of the HRUs (are they similar in size)? Can you include HRUs boundaries on Fig. 3?

**Response #5**

The spatial extent of the "RAF" HRU corresponds to the accumulation areas identified by the JFLOW simulation. Those of the others is determined by the Topographic Wetness Index (TWI, see later comment) as for TOPMODEL. The HRU boundaries can be included on the diagram as requested. The figure will be revised to show HRU boundaries and their areas given. The areas range from 2 to 101 km$^2$.

**Comment #6**

Figures need to be improved. The lines/dots have similar colours (eg. fig 5b - orange, pink). Yellow lines are not visible (fig. 6). Corresponding plots (fig. 4 and 6) have different starting and ending dates, precipitation is not consistent. Etc.

**Response #6**

The yellow lines in have been darkened to make clearer (see Figure 4). Consistent time intervals have been applied to all plots. Colours in the GLUE plots (Figure 3) have been revised to make clearer.
**Comment #7**

I don't understand to the description of the GLUE results (p.13, l. 13). Why the Ac around 65% gives the best NSE? It does not seem like that on Fig. 5.

**Response #7**

Due to the problem with the colouring (see response to comment 19) the Ac with the best NSE wasn't identified: it is in fact 75%.

**Comment #8**

In abstract and introduction you stress out that the unfortunate synchronization of the RAFs overtopping may cause serious problems downstream, therefore the RAFs network must be designed wisely. I do not see how you solved the desynchronization problem in the Results (slightly mentioned in Discussion). Can you please address this issue in more details in Results, Discussion and Conclusion?

**Response #8**

This is a potentially significant effect but the paper does not seek to address it as such, rather to provide a computationally-efficient modelling strategy that will allow investigation into the effect of hillslope storage on network flood wave timing. We will attempt to expand on this in the Results, Discussion and Conclusion sections.

**Comment #9**

There are many typos and grammatical mistakes in the text, units often follow the number without added space in between (eg. p.4, l. 13-14: 27000m3, 9%). Cited authors in the text sometimes do not agree with the list of references - check the years and Author's spelling please (eg. p. 4, l. 20: Pattison or Pattinson, l. 13: Ghmire, p. 10 l. 9: Binley  Beven should be Beven  Binley?, etc.). Sometimes decimal separator for thousands is used (p.4, l. 17), mostly not. p.1

**Response #9**
Thank you for pointing these out, we will endeavour to eliminate the mistakes.

**Comment #10**

Abstract: The abstract is very well written, I enjoyed reading it! Only the last sentence is misleading. At the moment "ways in which features could be grouped more strategically" are not given in the manuscript (there are general hints which are not supported by the simulation results). This sentence should be excluded or (preferably) the information included in the manuscript.

**Response #10**

We shall reword or remove this sentence. The ability to group features better (spatially and / or according to their morphology and topographic context) would improve the fidelity of the model but although discussed has not been undertaken in this study. It will make up part of future work.

**Comment #11**

p.2, l. 25: If JFLOW analysis and the workshop were used for this study, the information should be moved to Methods, rather than to state it in Introduction. The workshop and its outcomes are not further mentioned.

**Response #11**

The JFLOW analysis was presented at the workshop in order to select subcatchments for detailed modelling and refine the location of the enhanced hillslope storage, so this, as the reviewer points out, belongs in Methods rather than the discussion.

**Comment #12**

p.2, l. 30: The sentence repeats the information from the previous sentence (as RAF is also flood mitigation measure).

**Response #12**

Will be removed.

**Comment #13**

P3, l. 11: word repetition (would) p.4, l. 14: word repetition (in)

**Response #13**

Will be corrected.

**Comment #14**

p. 5, l. 22: Can you please specify why subsurface routing allows more flexible HRU aggregation in this case? How is it related to this study where the surface runoff is of the primary interest? Is this information related to the following sentence "Of particular relevance. . ."?

**Response #14**

In the original TOPMODEL the response units must be defined strictly in terms of the TWI in order to establish a relationship between the storage deficit in individual units relative to the mean deficit over the catchment. In the later Dynamic model, any landscape characteristic may be used as the routing is undertaken explicitly with a kinematic wave formulation. In the case study the TWI was applied to subdivide the catchment, but the RAF areas were then introduced as an additional unit overriding any previous landscape classification. This allowed simulation of the effect on the storm runoff of these areas intercepting overland flow redistributed from upslope areas. This would not have been possible in TOPMODEL.

**Comment #15**

p.7, l. 29: Which "study case"? The case study area has not been introduced yet. The chapter 2.3 mixes introduction and methods together.

**Response #15**

These ambiguities will be addressed.

**Comment #16**

p. 8: Can you justify or discuss how multiple RAFs are lumped into single RU? I don't think that the weir equations are necessary to be presented here.

**Response #16**

The equation are not necessary, it is true, and can be removed. The lumping of RAFs into a single unit is clearly a radical simplification that averages across all features the distribution of upslope input and downslope outputs, and applies the same "leakiness" and maximum storage capacity to each. It is argued later, however, that a much more fine-grained classification system could easily be applied, with the limiting case where each feature is associated with a single HRU. We shall attempt to make this clearer at this stage in the paper.

**Comment #17**

p.11, Study site: describe the instrumentation that provides the validation data, please. How the discharge is monitored (what kind of flume/weir, what capacity)? P.11, l 22. - the other gauge should be already mentioned together with the GM Bridge gauge.

**Response #17**

These details shall be included.

**Comment #18**

p. 12, l. 20: can you also include the lambda factor values?

**Response #18**

OK.

**Comment #19**

p. 13, l. 10: 7 distinct values of Ac are mentioned in the text, only 3 are on Fig. 5.

**Response #19**

Unique colours were not being applied to the unique values. This has been rectified: see Figure 3.

**Comment #20**

p. 13, l. 18: This paragraph belongs to Methods

**Response #20**

Will be moved.

**Comment #21**

Some literature is missing in the references: Dadson et al. (2017), Beven  Blazkova (2004), EA (2009), Marsh (2016), Beven  Wood (1983), Chappell et al. (2006)

**Response #21**

Thank you for bringing these omissions to our attention; they will be addressed.

**Comment #22**

Fig. 3: I suggest to include the HRUs boundaries. Position of the NFMs would be also interesting, but there are maybe too many of them.

**Response #22**

As for Comment 4.  It may be that putting both the HRU boundaries and the RAF positions is confusing. Our inclination would be to show only the RAFs. We include a revised figure (Figure 1) with these boundaries included.

**Comment #23**

Fig. 4: Why are there more red lines? I would think that there is only one observation

in the single gauge.

**Response #23**

Thank you for pointing this out. The display of the observed values are indeed in error as the colours had been incorrectly applied; every other time series was being displayed in red. This has been fixed: see Figure 2.

**Comment #24**

The Precipitation bars are too thick. In the given scale the daily precipitation amount on 13th Nov would be app 240 mm.

**Response #24**

This has been addressed: see Figures 2 and 4.

**Comment #25**

Fig. 5: What are the thin lines with max excess storage around 0.2 m, visible after Desmond storm? Why are they so different from most of the simulation results (for all the RAFs)?

**Response #25**

These were the results of simulations that encountered numerical errors and should have not been included in the set identified as behavioural. We will try to pin down what is causing this anomalous behaviour. The corrected figure is included here as Figure 4.

**Fig. 1.** Eden headwaters to Great Musgrave Bridge (223km$^2$), showing context within Cumbria, UK, rain gauges and gauging stations, and hillslope storage sites identified through JFLOW screening.

**Fig. 2.** Simulated discharges at Great Musgrave Bridge across the calibration period described in the main text for behavioural realisations alongside rated observed discharges.

**Fig. 3.** GLUE "dotty" plots showing overall weighting (likelihood) scores for each of the 348 behavioural runoff simulations identified against the three model outputs described in the text.

Surface storage equivalent (mm)

Precip. (mm/hr)

02-Nov 04-Nov 06-Nov 08-Nov 10-Nov 12-Nov 14-Nov 16-Nov 18-Nov 20-Nov 22-Nov 24-Nov 26-Nov 28-Nov 30-Nov 02-Dec 04-Dec 06-Dec 08-Dec 10-Dec 12-Dec 14-Dec 16-Dec

—— RAF1    —— RAF10    —— RAF100

**Fig. 4.** Surface excess storages across the lumped RAF unit with maximum storage set to 1m, applying each of the three mean residence time scenarios considered.

---

## Author Response (AR1)

**Reviewer 1**

Comment #1

The term "runoff attenuation features" is establishing itself in the literature but it is an C1 HESSD Interactive comment unfortunate use of overcomplicated language where simple language will do. In some cases RAFs are simply what most people

5 would call ponds! In other cases the use of the term obscures the relative effects of storage and attenuation and glosses over the variety of interventions that might be possible by lumping them together as RAFs. My suggestion is to be clear about which types of intervention are being referred to at the outset, and to prefer simple language if possible.

We agree that the lumped term can be misleading or inaccurate and could be considered unnecessarily technical. Our suggestion is to use "enhanced hillslope storage" (EHS) or improved or additional hillslope storage

10 (features).

Comment #2

Some clarity on how the drainage time constant is estimated (p3. Line 5) is essential. It appears to be allowed to range over two orders of magnitude for the purposes of the uncertainty estimation exercise. Fine if that's all that can be done but some discussion of what values the parameter might sensibly take is warranted in a paper of this kind. Is it considered an observable

15 parameter that an engineer might design to or control?

We suggest that that the drainage time could vary by this order of magnitude according to the feature type in question: earth bunds will drain down many orders of magnitude than leaky dams. However, we qualify this in the by observing that these values are not empirically determined, and suitable values are chosen to reflect the large range anticipated in these times.

20 Comment #3

Given the simplicity of the model there is an overly confident equation of model performance with the operation of the "real world". For example on p. 3 line 20 after only just having introduced the modelling approach it is declared that the model "can be used to examine the drain down, filling and possible overflow of these features during the course of storm events." I understand the point that's being made but it would be as well to note that

25 the model can only provide an understanding commensurate with the fidelity of its representation of the interventions.

We agree is true and such statements have been removed or qualified where they occur.

Comment #4

The opportunity on p5 line 20 to describe whether different types of feature behave as modelled is also missed (and also on p6

30 line 25 when it is stated that tree shelterbelts can be modelled using this approach too). I suggest that some additional text in these parts of the manuscript would help to improve the paper.

We have removed the example of shelterbelts as irrelevant.

Comment #5

A key part of the research problem is to distinguish between storage effects and network / wave propagation effects. This is

35 very clearly articulated on p4 lines 21ff, an explanation that might more helpfully be given earlier in the paper. The comment

p17 line 22 that network effects are important is appropriate to the discussion (and has been raised in the literature before) but it is hard to see what new evidence is offered for this view in the present manuscript.

Although the paper in itself only circumstantial evidence, it is suggested that due to its simplicity is could be usefully applied to an "experimental" modelling approach to investigating the Synchronisation effect. A statement to this effect has been added to the Conclusions ()

Comment #6

The tables indicate the results of the study very clearly but the text is a little more equivocal, and is unclear in parts. For example, it would appear from the tables that the RAF10 intervention shows a 30% reduction in peak flow – an important effect - but this is not discussed until the conclusions and is only mentioned in passing (p16 line 32). More detailed presentation of the results in Section 5 is warranted, in my view.

The results originally presented were not likelihood weighted, which gave a false impression of the magnitude of the impacts. The data have been recomputed so that each difference between simulated peak discharges for base and intervention cases is likelihood-weighted, by multiplying the value by the GLUE likelihood score (shown in Figure 5). The weighted maximum impact for RT10 is now 17.3% through Storm Desmond. The caption to Table 3 has been expanded with this information. The discussion of the results (pp11-13) expands on this.

Comment #7

Minor comments There are a number of typos / grammatical mistakes / errors of punctuation throughout, which might usefully be corrected at this stage.

We are grateful for the reviewer pointing this out and we have endeavoured to eliminate all of these errors.

Comment #8

The title of the paper somewhat understates the conclusions of the piece. In its current form the title suggests that the paper offers only an improvement in method. No doubt an improvement in the simulation of runoff attenuation features would be worthwhile in its own right but the work presented takes the ideas further and in fact evaluates and seeks to draw conclusions on the performance of systems of RAFs under multiple storm conditions. I'd suggest that the title be edited to reflect the wider scope of the paper as written.

Our revised title is "A new method, with application, for analysis of the impacts on storm runoff and flood risk of widely-distributed enhanced hillslope storage"

Comment #9

Please clarify the statement at p14 line 6: "Due to hydrodynamic storage utilisation exceeds 100% at the peak of Storm Desmond. . ."

The weir equation applied to model overflow out of the features will result in a small head clear of the rim that will introduce more storage over and above the hydrostatic amount. This storage cannot be identified beforehand as its is not known how large this head will be. Hence, the static storage of the feature at "brim-full" is equated with 100% utilisation. Clarified p 12, line 8.

Comment #10

Table 4. Consider the implied precision of the modelled estimates when quoting simulated peak delays in hours to two decimal places.

These have been given to 1 d.p.

**Reviewer 2**

Comment #1

I must admit I have some difficulties to follow the text in the first chapters (p. 2-12). Structure of the manuscript is a little bit confusing; it does not follow the classical IMRAD approach. Same information are repeated on several places, paragraphs are

5   sometimes too long containing too many topics. Often I am not sure if the authors describe actual research or results from the previous studies (referred as Metcalfe at al., or Hankin et al.). Simplification of the text structure and rephrasing of long sentences would very much help to increase the readability and overall appeal.

The introductory sections pp1-12 have been comprehensively revised according to these suggestions, in content and structure.

10   Comment #2

p.3, l. 9: I don't clearly understand which PROJECT you refer to. Is it related to Hankin et al. (2017) or another previous work? Or do you address actual research presented in this manuscript? Similarly, on the rest of p. 3, where several times "objectives of the project" are mentioned. At the moment the text reminds me more of a project proposal than an article. My suggestion is to move most of the chapter 1.1 directly to the Introduction. In Aims and Objectives simply state the "aims, objectives,

15   hypotheses" (even the bullets will make it). I would also omit the hints of the used methods in Introduction (eg. different levels of leakiness through the walls, Dynamic TOPMODEL, Storm Desmond etc.) and leave it for the following chapters. Just to make the text easier to follow.

The "project" was the Rivers Trust IP project undertaken August – November 2016. The aims and objectives of this project have been condensed to a few paragraphs p4, l32 – p5, l14, and linked to the study presented.

20   Comment #3

p. 4, ch 1.2 – I suggest to move the whole chapter before Aims and Objectives. This is still pure introduction.

The introduction has been largely rewritten. The suggested text has been moved from the methods section into the introduction. The methods section has been renamed as such and is concentrated on describing the development of the modelling approach applied to the study.

25   Comment #4

p. 7, 1st paragraph: it was already stated before that distributed modelling is computationally demanding. You have already introduced the simplified approach that you use. This paragraph should be a part of Introduction.

The comment has been removed.

Comment #5

30   p. 12 I don't understand how 8 HRU units correspond to the JFLOW simulation. Was JFLOW used on 8 separate subregions? Or are the HRUs related to the TOPMODEL simulation? Please, clarify. What is approximate size of the HRUs (are they similar in size)? Can you include HRUs boundaries on Fig. 3?

The approach has been clarified in the opening of section 3.1 p 10, l14-19. JFLOW was used in an initial "screening" stage (across the entire catchment) in order to identify areas that could be deepened to provide

additional runoff storage. Subsequently all analysis was undertaken in Dynamic TOPMODEL. The spatial extent of the "Hillslope Storage" HRU corresponds to the accumulation areas identified.

The other HRUs range in size from 101 km$^2$ to 2 km$^2$; added (p10, l)

Comment #5

5 Figures need to be improved. The lines/dots have similar colours (eg. fig 5b – orange, pink). Yellow lines are not visible (fig. 6). Corresponding plots (fig. 4 &6) have different starting and ending dates, precipitation is not consistent. Etc.

The yellow lines in Figure 5 have been darkened to make clearer. Consistent time intervals have been applied to all plots. Colours for Figure 6 have been revised to make clearer.

Comment #6

10 I don't understand to the description of the GLUE results (p.13, l. 13). Why the Ac around 65% gives the best NSE? It does not seem like that on Fig. 5.

Due to the problem with the colouring (see response to comment on numbers of distinct values) the Ac with the best NSE wasn't identified: it is in fact 75%

Comment #7

15 In abstract and introduction you stress out that the unfortunate synchronization of the RAFs overtopping may cause serious problems downstream, therefore the RAFs network must be designed wisely. I do not see how you solved the desynchronization problem in the Results (slightly mentioned in Discussion). Can you please address this issue in more details in Results, Discussion and Conclusion?

This is a potentially significant effect but the paper does not seek to address it as such, rather to provide a
20 computationally-efficient modelling strategy that will allow investigation into the effect of hillslope storage on network flood wave timing.

Comment #8

There are many typos and grammatical mistakes in the text

Units often follow the number without added space in between (eg. p.4, l. 13-14: 27000m3, 9%).

25 Cited authors in the text sometimes do not agree with the list of references - check the years and Author's spelling please (eg. p. 4, l. 20: Pattison or Pattinson, l. 13: Ghmire, p. 10 l. 9: Binley & Beven should be Beven & Binley?, etc.).

Sometimes decimal separator for thousands is used (p.4, l. 17), mostly not. p.1

Thank you for pointing these out, we have endeavoured to eliminate the mistakes. The decimal separator is not now used for powers of ten greater than 4 (e.g. 1000, 10,000, 100,000 etc)

30 Comment #9

Abstract: The abstract is very well written, I enjoyed reading it! Only the last sentence is misleading. At the moment "ways in which features could be grouped more strategically" are not given in the manuscript (there are general hints which are not

supported by the simulation results). This sentence should be excluded or (preferably) the information included in the manuscript.

Thanks you. We have reworded this final sentence. The ability to group features better (spatially and / or according to their morphology and topographic context) would improve the fidelity of the model but although discussed has not been undertaken in this study. It will make up part of future work.

Comment #9

p.2, l. 25: If JFLOW analysis and the workshop were used for this study, the information should be moved to Methods, rather than to state it in Introduction. The workshop and its outcomes are not further mentioned.

The JFLOW analysis was presented at the workshop in order to select subcatchments for detailed modelling and refine the location of the enhanced hillslope storage, so this, as the reviewer points out, belongs in Methods rather than the discussion. It is mentioned only briefly at the start of section

Comment #9

p.2, l. 30: The sentence repeats the information from the previous sentence (as RAF is also flood mitigation measure).

This has been removed.

Comment #10

P3, l. 11: word repetition (would) p.4, l. 14: word repetition (in)

Corrected

Comment #11

p. 5, l. 22: Can you please specify why subsurface routing allows more flexible HRU aggregation in this case? How is it related to this study where the surface runoff is of the primary interest? Is this information related to the following sentence "Of particular relevance. . ."?

In the original TOPMODEL the response units must be defined strictly in terms of the TWI in order to establish a relationship between the storage deficit in individual units relative to the mean deficit over the catchment. This was a consequence of the quasi-semi state water table assumption made in the model.

In the later Dynamic model, the subsurface routing is undertaken explicitly with a kinematic wave formulation. This allows any landscape characteristic to be used as a basis to aggregate HRUs. In this situation it allows areas associated with hillslope NFM interventions to be identified with one or more units.

Comment #11

p.7, l. 29: Which "study case"? The case study area has not been introduced yet. The chapter 2.3 mixes introduction and methods together.

These ambiguities has been addressed.

Comment #12

p. 8: Can you justify or discuss how multiple RAFs are lumped into single RU? I don't think that the weir equations are necessary to be presented here.

Given the assumption of a depth, sThe unit behaves as one large pond whose drainage area (the walls through which stored runoff leaves) scales linearly with the volume. Thus the discharge only depends on specific storage.

The Weir equation does not need to be supplied as given in any hydraulic textbook, and has been removed.

The lumping into a single unit is clearly a radical simplification that averages across all features the distribution of upslope input and downslope outputs, and applies the same "leakiness" and maximum storage capacity to each. It is argued later, however, that a much more fine-grained classification system could easily be applied, with the limiting case where each feature is associated with a single HRU. We shall attempt to make this clearer at this stage in the paper.

Comment #12

p.11, Study site: describe the instrumentation that provides the validation data, please. How the discharge is monitored (what kind of flume/weir, what capacity)? P.11, l 22. – the other gauge should be already mentioned together with the GM Bridge gauge.

These details have been included.

Comment #12

p. 12, l. 20: can you also include the lambda factor values?

OK. Note that as the residence times are linked to the lambda value with the surface routing, the residence time was felt to be more relevant to exploring the effect of the additional storage. The intervention cases have been renamed RT1, RT1 and RT100

Comment #12

p. 13, l. 10: 7 distinct values of Ac are mentioned in the text, only 3 are on Fig. 5.

Unique colours were not being applied to the unique values. This has been rectified: see attached.

Comment #13

p. 13, l. 18: This paragraph belongs to Methods

Has been moved.

Comment #14

Some literature is missing in the references: Dadson et al. (2017), Beven & Blazkova (2004), EA (2009), Marsh (2016), Beven & Wood (1983), Chapell et al. (2006)

Chappell et al. and Beven and Wood removed. Reference to Luscombe et al., 2015 added to reflect advances in thermal imaging for estimation of saturated areas.

Thank you for bringing this to our attention, these mistakes have been addressed.

Comment #15

Fig. 3: I suggest to include the HRUs boundaries. Position of the NFMs would be also interesting, but there are maybe too many of them.

Showing both the HRU boundaries and the features positions produced a confusing diagram so we have retained the original diagram.

Comment #16

Fig. 4: Why are there more red lines? I would think that there is only one observation in the single gauge. The Precipitation
5    bars are too thick. In the given scale the daily precipitation amount on 13th Nov would be app 240 mm.

Thank you for pointing this out. The display of the observed values are indeed in error as the colours had been incorrectly applied; every other time series was being displayed in red. This has been addressed.

Comment #17

10   Fig. 5: What are the thin lines with max excess storage around 0.2 m, visible after Desmond storm? Why are they so different from most of the simulation results (for all the RAFs)?

These were the results of simulations that encountered numerical errors and should have not been included in the set identified as behavioural. We will try to identify what is causing this anomalous behaviour.

[revised manuscript text omitted]

**Figures**

[Figure]

**Figure 1. Work flow diagram for Monte Carlo simulation of storm runoff, and selection and weighting of behavioural realisations and application of NFM scenarios for forward prediction of change. The weight of lines leading from acceptable simulations reflects the weighting likelihood score in the validity of that realisation.**

**Carlo simulatio**
**behavioural rea**
**forward predict**
**acceptable simu**
**the validity of th**

[Figure]

**Figure 2. Hydrodynamic accumulation areas within Eden, identified by JFLOW analysis for a designed storm of return period of 30 years (Hankin et al., 2017). Maximum water depths are indicated, and areas that exceed the threshold depth and other criteria (minimum area, slope angle and proximity to roads and buildings) are highlighted as potential sites for EHS.**

¶

¶

Figure 1: Hydrody...
identified by JFLO...
period of 30 years...
are indicated, and...
other criteria (mi...
roads and buildin...

Figure 1: Hydrody...
identified by JFLO...
period of 30 years...
are indicated, and...
other criteria (mi...
roads and buildin...

[Figure]

Figure 3. Study catchment, the Eden headwaters to Great Musgrave Bridge (223km²), showing context within Cumbria, UK, predominant land cover and location of TBR rain gauges and gauging stations. Woodlands for Water tree planting opportunity areas are shown. These were applied in another application of the NFM modelling framework developed for the project

¶

[Figure]

**Figure 4. Simulated discharges across the storm period alongside rated observed discharges at Great Musgrave Bridge. The three named storms are indicated. Rainfall is interpolated between the gauges shown in Figure 3.**

Figure 4. Simulate
across the calibrat
behavioural realis
The three named s
between the gauge

[Figure]

**Figure 5.** GLUE "dotty" plots showing overall likelihood (weighting) scores calculated for the 348 behavioural runoff simulations against the three model outputs described in the text. The discontinuous appearance of the maximum saturated contributing area $A_c$ is due to the relatively coarse discretisation applied such that once a HRU begins to produce any saturated overland flow, its entire area is added. Each unique $A_c$ value takes a separate colour that is carried through to corresponding points in the other plots.

| Deleted: 5 |
| Deleted: weighti |
| Deleted: ) |
| Deleted: for eac identified |
| Deleted: |

[Figure]

[Figure]

Figure 6. **Surface excess storages, expressed as specific rainfall equivalent, across one of the lumped RAF units with maximum storage 1m through a single intervention cases and for the three mean residences times considered. The slight excess at the peaks of the storm reflects the weir crest height of the overflow function applied.**

[Figure]

**Figure 7.** (L) 90th percentile likelihood scored baseline and corresponding intervention cases through Storm Abigail. (R) Likelihood-weighted cumulative frequency plot of peak discharges for base and intervention cases. Note that, in order to share the same vertical axis,  the Cumulative frequency plot is transposed relative to convention.

[Figure]

**Figure 8.** **(L) 90th percentile scored** base line and corresponding intervention cases through Storm Barney. **(R) Cumulative frequency plot of peak discharges for base and intervention cases.**

[Figure]

**Figure 9.** (L) 90th percentile scored base line and corresponding intervention cases through Storm Desmond. (R) Cumulative frequency plot of peak discharges for base and intervention cases.

¶
¶
¶

**Figure 3. Study ca
Musgrave Bridge
UK, predominant
gauges and gaugin
Woodlands for Wa
applied in another
developed for the
detail here.¶**
¶
**Figure 1: Hydrody
headwater catchm
designed storm eve
water depths are i
depth and other cr
proximity to roads
sites for RAFs.¶
Figure 2: Suggeste
storm runoff, and
realisations and ap
prediction of chan
acceptable simulat
the validity of that
Figure 3. Study ca
Musgrave Bridge
UK, predominant
gauges and gaugin
Woodlands for Wa
applied in another**

**Moved up [8]:** F
**Musgrave Bridge
UK, predominant
gauges and gaugin
Woodlands for Wa**

**Moved (insertic**

---

## Author Response (AR2)

**Responses to reviewers**

| | |
|---|---|
| **Simon Dadson, simon.dadson@ouce.ox.ac.uk**

The authors have revised this manuscript carefully taking account of all the comments made. The additional clarity with which results in Table 3 and Section 5 are presented is very welcome. | |
| In relation to the revised description of Storm Desmond there is a very recent paper here worth citing: T Matthews et al 2018 | **Thank you for drawing our attention to this paper; a reference to it has been added (p9).** |
| **Anonymous referee**

In my opinion the major comments were addressed, the raised issues clarified. I do not have any more comments. I recommend to accept the manuscript once the minor editing according to the HESS standards is done (please see the Guidelines for the authors): | |
| Spaces must be included between number and unit (e.g. 1 %, 1 m)." Especially check on p. 1 and 9. | **These have been corrected where they occur.** |
| Units must be written exponentially. Correct please in Tables 1&2 (column Units) and on axis descriptions on the Figs. 4-9. | **Any occurrences have been modified as requested.** |
| Common abbreviations to be applied: hour as h (not hr)" Correct please in Section 2.2 and in the Figures. | **Modified as requested throughout** |
| - p. 7, l. 29. Correct the newly added sentence "The units representing the are ..."
p. 9, l. 3. Typo "OModifications"
p. 9, l. 25. 372 m3 s-1 (unit) | **These errors have been corrected.** |

[revised manuscript text omitted]

**Figures**

[Figure]

**Figure 1. Work flow diagram for Monte Carlo simulation of storm runoff, and selection and weighting of behavioural realisations and application of NFM scenarios for forward prediction of change. The weight of lines leading from acceptable simulations reflects the weighting likelihood score in the validity of that realisation.**

[Figure]

**Figure 2. Hydrodynamic accumulation areas within Eden, identified by JFLOW analysis for a designed storm of return period of 30 years (Hankin et al., 2017). Maximum water depths are indicated, and areas that exceed the threshold depth and other criteria (minimum area, slope angle and proximity to roads and buildings) are highlighted as potential sites for EHS.**

[Figure]

**Figure 3. Study catchment, the Eden headwaters to Great Musgrave Bridge (223 km²), showing context within Cumbria, UK, predominant land cover and location of TBR rain gauges and gauging stations. Woodlands for Water tree planting opportunity areas are shown. These were applied in another application of the NFM modelling framework developed for the project described, which are not discussed in detail here.**

[Figure]

**Figure 4. Simulated discharges across the storm period alongside rated observed discharges at Great Musgrave Bridge. The three named storms are indicated. Rainfall is interpolated between the gauges shown in Figure 3.**

[Figure]

**Figure 5. GLUE "dotty" plots showing overall likelihood (weighting) scores calculated for the 348 behavioural runoff simulations against the three model outputs described in the text. The discontinuous appearance of the maximum saturated contributing area $A_c$ is due to the relatively coarse discretisation applied such that once a HRU begins to produce any saturated overland flow, its entire area is added. Each unique $A_c$ value takes a separate colour that is carried through to corresponding points in the other plots.**

¶

[Figure]

**Figure 6. Surface excess storages, expressed as specific rainfall equivalent, across one of the lumped RAF units with maximum storage 1 m through a single intervention cases and for the three mean residences times considered. The slight excess at the peaks of the storm reflects the weir crest height of the overflow function applied.**

[Figure]

**Figure 7. (L)** 90th percentile likelihood scored baseline and corresponding intervention cases through Storm Abigail. **(R)** Likelihood-weighted cumulative frequency plot of peak discharges for base and intervention cases. Note that, in order to share the same vertical axis, the Cumulative frequency plot is transposed relative to convention.

[Figure]

**Figure 8. (L) 90th percentile scored base line and corresponding intervention cases through Storm Barney. (R) Cumulative frequency plot of peak discharges for base and intervention cases.**

[Figure]

**Figure 9. (L) 90th percentile scored base line and corresponding intervention cases through Storm Desmond. (R) Cumulative frequency plot of peak discharges for base and intervention cases.**